# Aberrant sorting of hippocampal complex pyramidal cells in type I lissencephaly alters topological innervation

**James A D'Amour[1,2], Tyler Ekins[1,3], Stuti Ganatra[1], Xiaoqing Yuan[1], Chris J McBain[1]\***

[1]Program in Developmental Neurobiology, Eunice Kennedy-Shriver National Institute of Child Health and Human Development, National Institutes of Health, Bethesda, United States; [2]Postdoctoral Research Associate Training Program, National Institute of General Medical Sciences, Bethesda, United States; [3]Brown University, Department of Neuroscience, Providence, United States

**Abstract** Layering has been a long-appreciated feature of higher order mammalian brain structures but the extent to which it plays an instructive role in synaptic specification remains unknown. Here we examine the formation of synaptic circuitry under cellular heterotopia in hippocampal CA1, using a mouse model of the human neurodevelopmental disorder Type I Lissencephaly. We identify calbindin-expressing principal cells which are mispositioned under cellular heterotopia. Ectopic calbindin-expressing principal cells develop relatively normal morphological features and stunted intrinsic physiological features. Regarding network development, a connectivity preference for cholecystokinin-expressing interneurons to target calbindin-expressing principal cells is diminished. Moreover, in vitro gamma oscillatory activity is less synchronous across heterotopic bands and mutants are less responsive to pharmacological inhibition of cholecystokinin-containing interneurons. This study will aid not only in our understanding of how cellular networks form but highlight vulnerable cellular circuit motifs that might be generalized across disease states.

**\*For correspondence:**
mcbainc@mail.nih.gov

**Competing interests:** The authors declare that no competing interests exist.

## Introduction

Cellular layers and refined somatic positioning are the hallmark of more evolutionarily developed brain structures. However, little is known regarding the contributions of layers to cellular maturation and local microcircuit formation. Disorders affecting cellular lamination offer a unique window of study into cellular and circuit development in the absence or disruption of traditional positional cues present in layers. Cellular heterotopias within brain structures result from a variety of developmental insults to an organism but share the common feature of lacking normal cellular layering typical in the cortex and hippocampus of mammalian brains; and like many things, how it falls apart speaks to its construction (*Anusha, 2014*; *Di Donato et al., 2017*).

While heterotopias may arise from diverse causes, they share some common phenotypes (*Kato, 2003*). Particularly devastating heterotopias involve mutations to genes that encode proteins essential to cellular migration and proliferation (*Hirotsune et al., 1998*). Brains from these patients often appear smooth, lacking the infoldings and gyri of healthy human subjects. Broadly, this condition is referred to as lissencephaly, meaning 'smooth brain'. One of the most common and first identified genetic causes of Type I lissencephaly is due to mutations in the Lis1 gene (*Pafah1b1*), which encodes an enzyme essential for nuclear kinesis and microtubule stabilization (*Hirotsune et al., 1998*; *Dobyns and Das, 2009*; *McManus et al., 2004*; *Wynshaw-Boris, 2007*). Unsurprisingly, mutations to other parts of this migratory pathway also result in lissencephalies and more recently

infections during embryonic development have received renewed attention for their role in microcephalies, such as the mosquito transmitted Zika virus (for example *DCX*, *YWHAE*, *RELN*, *ARX*) (*Kato, 2003*; *Rice et al., 2018*). As alluded to above, these disorders also produce intra-structure cellular heterotopias which are characterized by mispositioned cell somas and disorganized cellular layering.

Rodent brains lack gyri but mice heterozygous for the human mutant *Pafah1b1* allele (Lis1-MUT, Lis mutants) display severe cellular heterotopias in both cortex and hippocampus, developmental defects, hydrocephaly, and enlarged ventricles. These mice also have increased network excitability, lowered seizure threshold, and increased spontaneous mortality rate – features shared with the human condition (*Fleck et al., 2000*; *Hunt et al., 2012*). Interestingly, these heterotopias in area CA1 of the hippocampus have a tendency to fragment the single excitatory principal cell layer (PCL) into multiple pyramidal cell bands, stacked vertically on one another – transitioning the region into what looks like a primitive cortical structure with multiple excitatory layers. Concurrently, hippocampal researchers have proposed a system of parallel information processing being carried out among the intertwined circuitry of CA1, where-in preferential interneuron targeting acts to segregate information streams to different sets of principal neurons (*Soltesz and Losonczy, 2018*). It seems possible, if not likely, that these crude laminar structures resulting from faulty cellular migration in the Lis1 mutant mouse, might reflect natural underlying patterns in local circuit connectivity upon which normal hippocampal function is critically dependent. Clearly, mis-lamination is a shared feature of several human neurodevelopmental disorders that merits deeper investigation and may inform our understanding of normal hippocampal development and function.

In light of studies suggesting specified microcircuitry among deep versus superficial principal cells and local basket cells in wild type (Wt) CA1, we wondered if the heterotopic cell layers observed in Lis1 mutants reflected a functional distinction between discrete microcircuitry of the PCL (*Soltesz and Losonczy, 2018*; *Lee et al., 2014*; *Nielsen et al., 2010*; *Slomianka et al., 2011*; *Valero et al., 2015*). Recent evidence suggesting a preferential connectivity between principal cells and either parvalbumin (PV) or cholecystokinin (CCK) expressing interneurons, depending on the extrahippocampal projection target, somatic position of the principal cell, or marker expression of the principal cell, suggests an underlying blueprint in the establishment of hippocampal circuitry and connectivity that has been previously underappreciated in what otherwise appears as a monolithic excitatory lamina, the PCL (*Soltesz and Losonczy, 2018*; *Lee et al., 2014*; *Nielsen et al., 2010*; *Slomianka et al., 2011*; *Valero et al., 2015*; *DeFelipe, 1997*; *Deguchi et al., 2011*; *Valero and de la Prida, 2018*; *Varga et al., 2010*). This model will allow us to identify the same cell subtypes in non-mutant and mutant littermates and examine to what extent their relative relationships are preserved under mis-lamination. Put more succinctly, to what extent are innate wiring motifs disrupted under heterotopia?

Remarkably, in subjects suffering from cellular heterotopias that survive into adulthood cellular networks function surprisingly well and animals are often behaviorally indistinguishable from normal type littermates (*Salinger et al., 2003*; *Wagener et al., 2010*; *Wagener et al., 2016*). In the more thoroughly studied Reeler mouse model, that displays severe cortical and hippocampal mis-lamination, cells in cortex appear to be relatively healthy and are integrated into the local network (*Wagener et al., 2016*; *Boyle et al., 2011*; *Caviness and Sidman, 1973*; *Guy and Staiger, 2017*; *Polleux et al., 1998*). Collectively, the evidence suggests that the formation of functional synaptic connectivity has some innate resilience to mis-lamination and layers may play little to no role in the guidance and establishment of synaptic connectivity (*Wagener et al., 2016*; *Guy and Staiger, 2017*; *Caviness and Rakic, 1978*; *Guy et al., 2015*). Furthermore, if there was logic behind the dividing of these heterotopic cell populations the Lis1-MUT would represent an ideal model to assay the resilience of genetic network formation blueprints to the developmental/local-environment cues of intra-structure position and layering (*Soltesz and Losonczy, 2018*; *Harris and Shepherd, 2015*; *Margeta and Shen, 2010*; *Sur and Rubenstein, 2005*). This might permit us to determine over what relative distances genetic wiring programs are able to locate and synapse on the appropriate postsynaptic targets, shed light on what appears to be intertwined parallel circuitry for information processing in CA1, and identify synaptic connectivity motifs that are more susceptible to heterotopia than others (*Soltesz and Losonczy, 2018*; *Valero and de la Prida, 2018*; *Varga et al., 2010*). Ultimately, these studies provide key insight into what is the role of layers in higher mammalian brain structures and highlight the proper areas of study for future treatment of cellular heterotopias.

## Results

### Heterotopic banding of the principal cell layer in Lis1 mutant mice

Non-conditional *Pafah1b1* $^{+/-}$ mice were generated by breeding a *Pafah1b1* $^{+/\,Fl}$ line to *Sox2*-cre animals, heterozygous mice are referred to henceforth as Lis1-MUT. Lis1-MUT mice were often noticeably smaller than litter mates. Some animals displayed severe macroscopic brain abnormalities, including enlarged ventricles, hydrocephaly, intracranial bleeding, and spontaneous death. Lis1-MUT mice that survived to 3–5 weeks of age were used for experiments and subsequent breeding; non-mutant littermates were used as controls. In coronal sections from dorsal hippocampus Lis1-MUT animals displayed heterotopic banding of the principal cell layer (*Figure 1A*). Banding varied in severity, cell soma density, and septal-temporal extent. Most animals displayed the strongest banding in area CA1, with fewer mice showing multiple PCLs past region CA2. Region CA3 rarely appeared banded, but instead scattered and uncompacted. Mice occasionally had three distinct layers or clustered islands of cells, but most typically two prominent PCLs could be seen (*Figure 1A*, *right* vs *left*). Deeper bands were typically situated in what would be stratum oriens-alveus in a non-mutant animal. In measuring from the border of the alveus and the cortex radially (toward radiatum, known as the radial axis of CA1), the entirety of normal wild type (Wt) PCLs were located between ~175–300 µm. In Lis1-MUT mice, superficial bands were positioned between ~250–320 µm and deeper heterotopic bands (positioned closer to the alveus) were between ~100 and 190 µm. Of the two bands, the superficial tended to be more densely populated and closer to the normal positioning of the PCL in normal type mice (*Figure 1A and B*). We next considered whether these heterotopic bands were splitting randomly, or if the banding represented distinct cell populations.

### Calbindin expressing principal cells preferentially position in the deeper heterotopic band of CA1 in Lis1 mutants

In order to better understand the banding process in mutant mice, immunohistochemistry experiments were carried out for principal cell markers and quantified by normalized expression levels in each heterotopic band (*n* antibody stained cells/*n* dapi cells in same region of interest). In addition to marking a subpopulation of GABAergic cells, calbindin is expressed in superficial principal cells in several species (*Slomianka et al., 2011*). Consistent with these reports, our Lis1 wild-type litter mates show calbindin-expression among superficial principal cells of CA1 (*Figure 1B*, *left*). These cells are tightly packed, forming one-three rows of somas on the stratum radiatum adjacent (superficial) side of the PCL. Conversely, calbindin staining in Lis1-MUT mice showed a strong bias for calbindin-expressing principal cells to occupy the deeper heterotopic principal cell layer (*Figure 1B*, *right*). *Figure 1D* shows a normalized histogram of identified calbindin-positive cell soma positions in Lis1 mutants and wild-type litter mate controls. Note for quantification purposes, the deep and superficial bands are analyzed as separate regions of interest, numbers represent the fraction of cells in that particular band expressing calbindin. Analogously, the single wild-type PCL is divided in half radially (top versus bottom) and analyzed as separate deep and superficial bands (*Figure 1E*; Distal CA1 Wt: deep 8.9 ± 2.8%; superficial 25.1 ± 1.3%; Lis1-MUT: deep 18.0 ± 2.8%; superficial 4.4 ± 1.0%, n = 12 Wt and 12 Lis1-MUT slices from six animals, respectivel). This finding was not a general feature of having the Lis1 mutant allele, as in animals with less severe banding (or in the same slices nearer CA2) but still carrying the mutant *Pafah1b1* allele, the PCL displayed relatively normal, superficial calbindin soma positioning (*Figure 1F*; Proximal CA1 Wt: deep 16.3 ± 2.3%; superficial 30.0 ± 2.0%; Lis1-MUT: deep 12.1 ± 2.3%; superficial 19.0 ± 2.4%, n = 12 and 11, respectively). A large proportion of the principal cells expressing calbindin are being preferentially shifted into the deeper heterotopic band, but it should be noted that calbindin cells overall still represent a minority of the population in either location. Given that principal cells are generated near what becomes the alveus and migrate radially during embryonic development in a deep to superficial manner (*Caviness and Sidman, 1973*; *Angeivine, 1965*; *Stanfield and Cowan, 1979*), the calbindin staining pattern suggested a late born population undergoing migratory failure in the Lis1-MUT mouse.

### Embryonic development of the calbindin expressing principal cells

Superficial principal cells in normal mice arise near the end of gestation (Emb days 16–17) (*Slomianka et al., 2011*; *Caviness and Sidman, 1973*; *Angeivine, 1965*). Our initial data suggests

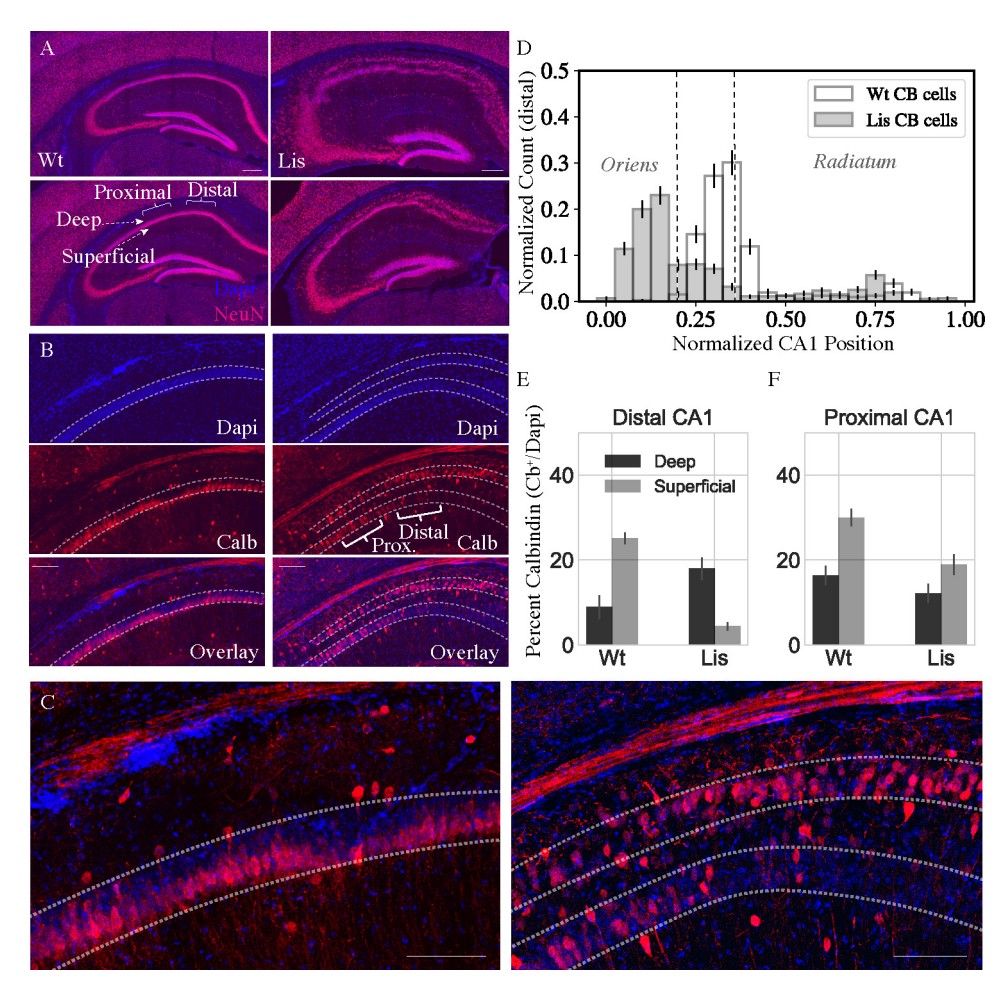

**Figure 1.** Lis mutants mice display heterotopic banding and ectopic positioning of calbindin-expressing principal cells. (A) *Left*, two coronal NeuN stained images from differing levels of dorsal CA1 hippocampus in a wild-type littermate. *Right*, approximately matched coronal sections from a Lis mutant displaying heterotopic banding of the PCL. Scale bars are 300 μm. (B) *Left*, wild-type and Lis1-MUT, *right*, staining of CA1 highlighting the position of the PCL, calbindin-positive neurons, and overlay. Note the deep layer preference of calbindin-expressing neurons, particularly in distal CA1 in mutant. Scale bars are 200 μm. (C) Higher magnification view of the overlay images in (B), for wild-type (*left*) and Lis1-MUT (*right*). Scale bars are 150 μm. (D) Normalized histogram showing the positioning of calbindin-expressing cells in mutants with PCL banding compared to wild-type mice. (E) Percentage of cells in deep and superficial layers expressing calbindin in distal CA1 (for controls mice, the single PCL is divided in half radially). Counts represent number of identified calbindin soma divided by number of DAPI identified cells, Wt: deep 8.9 ± 2.8%, superficial 25.1 ± 1.3%; Lis1-MUT: deep 18.0 ± 2.8%, superficial 4.4 ± 1.0%. (F) Same as (E) for proximal CA1, Wt: deep 16.3 ± 2.3%, superficial 30.0 ± 2.0%; Lis1-MUT: deep 12.1 ± 2.3%, superficial 19.0 ± 2.4%; n = 12 Wt and 12 Lis1-MUT slices for distal and 12 and 11 for proximal, from six animals. The online version of this article includes the following source data for figure 1:

**Source data 1.** Normalized calbindin cell counts within given regions of interest for distal and proximal CA1, normalized positional counts of CB+ cells.

that the heterotopic banding in Lis1-MUT mice may arise from a migratory stalling event, where later born superficial-preferring cells were unable to overcome a migratory burden and instead form a new deep heterotopic layer. In order to test this hypothesis and ensure that a novel population of deeply positioned principal cells was not adopting calbindin expression in Lis1-MUT animals, cellular birth-dating experiments were performed.

In timed pregnancy experiments using Lis1 mutants crossed to *Neurog2-cre* were crossed with a cre-dependent EGFP reporter mouse (Rosa26 <RCE > ), tamoxifen administration induces cre-

recombination and subsequent eGFP expression in newly born neurons of developing mouse pups. Pregnant mothers were gavaged at various embryonic time points spanning days E12-17. After pups were born, they were perfused and fixed at ~P30 for calbindin staining, and subsequent quantification of the percentage of eGFP expressing neurons from any time point that were co-stained for calbindin (*Figure 2A–C*). Approximately 10% of cells born on E12-E13 expressed calbindin at P30 (*Figure 2D*; Wt: 9 ± 3%; Lis1-MUT: 12 ± 3%, n = 95 cells and 168 labeled cells analyzed from five animals, respectively) in both Lis1 wild-type littermates and mutants. Cells born E14-E15 co-stained for calbindin 42 ± 9% of the time for wild type and 52 ± 8% (n = 221 and n = 128, from 10 animals) for Lis1 mutants and cells born E16-E17 co-stained for calbindin 54 ± 7% of the time for wild type and 71 ± 9% for Lis1 mutants (n = 48 and 20 labelled cells from 11 animals). While the timing of calbindin cell birthdates remained similar to littermate controls in Lis1-MUT animals in that calbindin cells arise late in embryonic development (*Figure 2D*), positioning of these cells differed substantially. Later born cells positioned more superficially in wild-type littermates (smaller PCL depth measurements), while they positioned more deeply in mutant mice (*Figure 2E and F,E* represents counts from single experiments data are averages and summarized in F). These results suggest that deeply positioned calbindin-expressing cells in the Lis1-MUT mice are the same late-born cell population that are now ectopically positioned in the deeper heterotopic band.

## Calbindin-expressing principal cells retain a complex apical morphology

Previous studies have documented variation in CA1 principal cell morphology, particularly in comparing basal and apical dendritic trees (*Lee et al., 2014*; *Bannister and Larkman, 1995*; *Li et al., 2017*). These morphological features can be reliably used to differentiate excitatory neuron subtypes. In particular, calbindin-expressing principal cells have more complex apical dendritic trees (more branching), than calbindin-negative counterparts (*Li et al., 2017*). This has enabled offline characterization of excitatory cell group through K-means clustering of morphological features after cellular reconstructions. A prior study using this approach suggested that clustering was greater than 90% accurate as verified by mRNA and in situ hybridization approaches but comes with the drawback that every recording must be histologically processed, virtually reconstructed, and analyzed (*Li et al., 2017*). Additionally, there is a minimal threshold for the amount of dendritic tree that must be recovered and drawn for clustering to be accurate.

In our 63 best recovered principal cell morphologies from physiological recording experiments (n = 32 wild type, n = 31 Lis1-MUT), we implemented a k-means clustering algorithm based on dendritic branch connectivity and lengths that generates length-ratio index (LRI) values and node-ratio index values (ORI), as done previously, with a minor modification to cell values if selected nodes were distant from the soma (*Li et al., 2017*). See Materials and methods for more details. The clustering results from mutant and wild-type litter mate cellular morphologies are shown in *Figure 3B*. The same process was applied to mutant and wild-type cells, but these groups were processed independently by a supervised k-means algorithm that expected two groups, corresponding to complex and simple morphologies. While morphologically speaking, additional subtypes of principal neurons likely exist in CA1 (if analyzing basal dendrites or soma size etc.), the present study makes use of prior work for the purposes of separating principal cells along the lines of calbindin positive and calbindin negative populations. However, preliminary clustering analysis using within cluster sum of squares (elbow plots) suggested ncluster = 2 is not an unreasonable choice for apical dendritic morphology in agreement with the original study (data not shown). Note, not all cells come from the same experimental group as in this figure we are looking for the best morphological recoveries, therefore not every morphological data point will have corresponding physiological data points causing N's to vary between some subpanels – the same should be noted for the analysis in *Figure 4*.

The data show that relatively simple and complex cell morphologies persist in the Lis1 mutant, in approximately the same proportions to wild type mice, with nearly overlapping cluster centers (complex cells, Wt: [−0.1, 0.8], Lis1-MUT: [−0.4, 0.9]; simple cells, Wt: [−1.8,–1.3], Lis1-MUT: [−1.7,–1.2]). The nearness of the cluster centers for non-mutant and mutant data further suggest that fundamentally new clusters of morphological subtypes have not cropped up, but instead a relative relationship between complex and simple subtypes persists – even if Lis1 complex cells are less 'branchy' as described below in sholl analysis. A visual comparison of some of the more obviously simple and complex cellular recoveries suggests the sorting has been successful (*Figure 3A & C*, note deeply

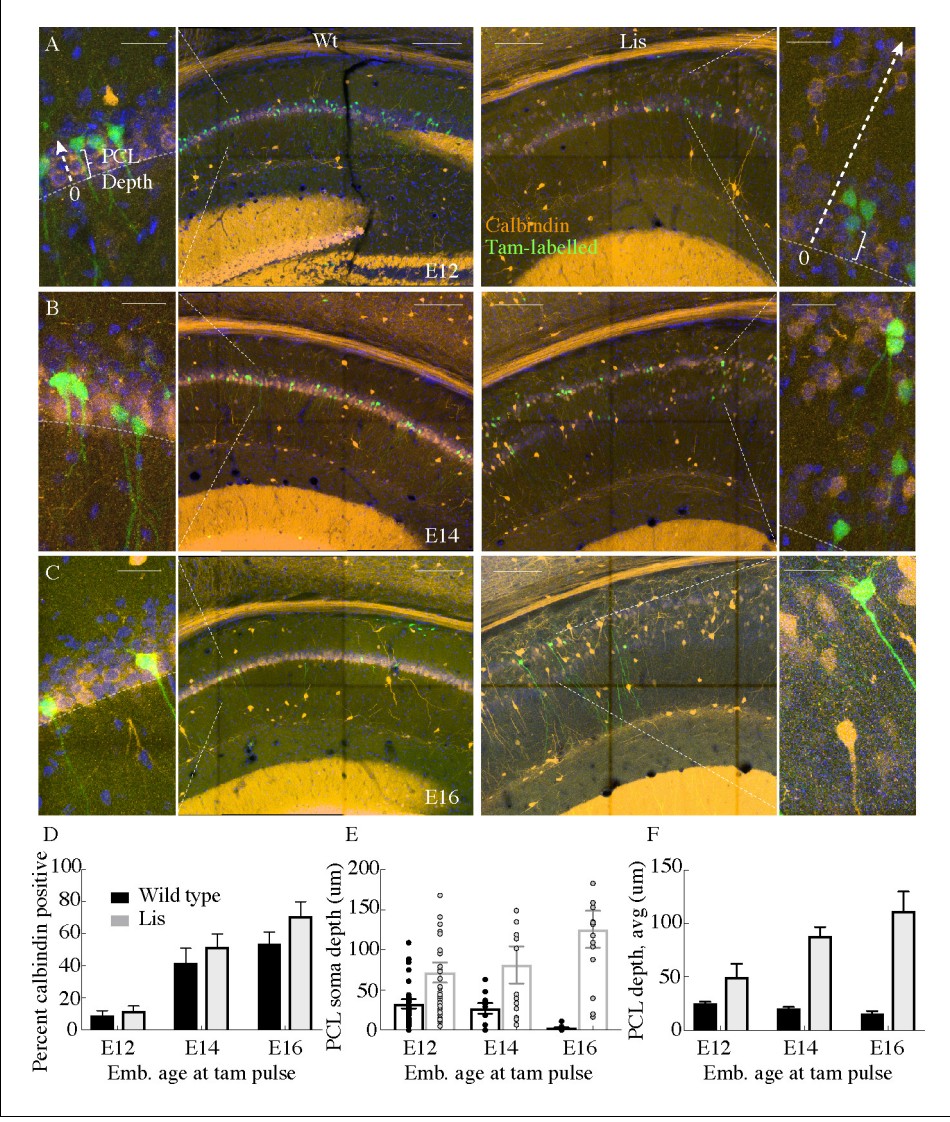

**Figure 2.** Cellular birth-dating indicates ectopic calbindin cells in Lis mutants are the same late-derived embryologic population. (**A**) Wild-type (*left*) and Lis mutant (*right*) example birth-dating images for a litter tamoxifen dosed between E12-E13. Note the cutout, displaying how cellular somatic positioning was measured from the front of the PCL (as opposed to normalized structural position). Green corresponds to cells born during tamoxifen administration; orange is calbindin immunohistochemistry staining carried out when litters are P30. (**B**) Same as in (**A**) but for litters dosed at E14-E15. (**C**) Same as (**A**) but for litters dosed at E16-E17. (**D**) Quantification of the fraction tamoxifen-marked neurons co-staining for calbindin antibody from each timepoint. E12: Wt: 9 ± 3%; Lis1-MUT: 12 ± 3%, E14: Wt: 42 ± 9%; Lis1-MUT: 52 ± 8%, E16: Wt: 54 ± 7%; Lis1-MUT: 71 ± 9%. (**E**) Example counts from single images at each timepoint for PCL depth measurements. Later born cells position more superficially (front of the PCL) in non-mutants, but deeper in Lis1-MUT littermates. (**F**) Group averages for the measurements shown in (**E**). E12- Wt: 25.4 ± 1.4 μm; Lis1-MUT: 50 ± 12.3 μm, E14- Wt: 20.5 ± 1.3 μm; Lis1-MUT: 88 ± 8.7 μm, E16-Wt: 15.6 ± 2.6 μm; Lis1-MUT ⁻: 111.6 ± 18.3 μm. Scale bars are 150 μm and 25 μm for the main image and zoom, respectively.

The online version of this article includes the following source data for figure 2:

**Source data 1.** Summary percentage CB+ cells with Tam labelling, PCL depth measurements for cells in single mouse examples, summary data with PCL depths.

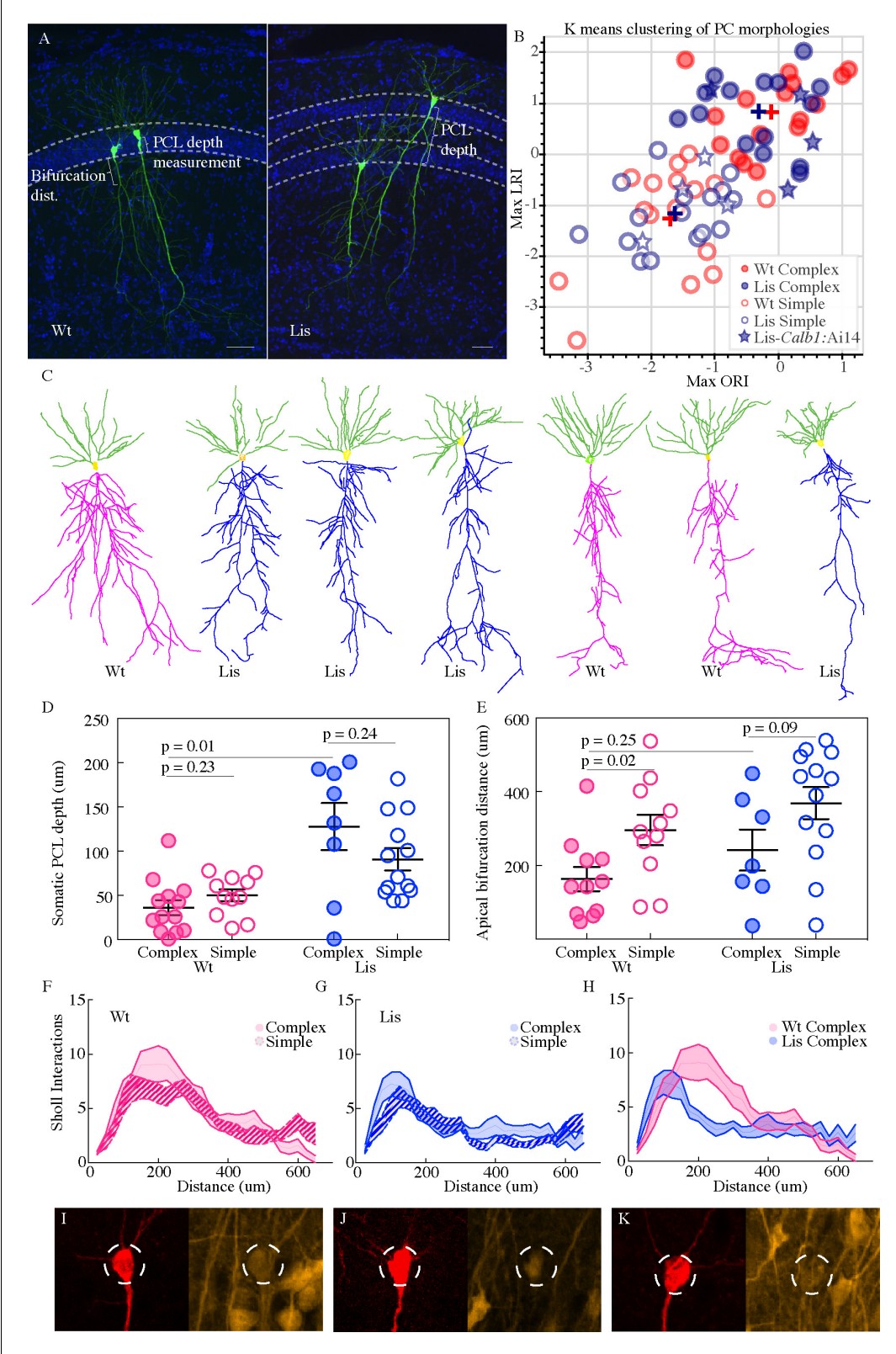

**Figure 3.** Lis1-MUT calbindin-expressing PCs retain relatively complex morphologies. (**A**) Recovered cells from non-mutant and mutant experiments, highlighting different apical dendritic morphologies, complex and simple. Complex morphologies have been previously shown to be highly predictive of calbindin expression (*Li et al., 2017*). Scale bars are 50 μm. (**B**) Supervised K-means plots (63 best recovered cells, k = 2) carried out separately for mutant and wild-type data (blue and red respectively). Filled circles correspond to complex morphologies and open circles are simple. Stars are

*Figure 3 continued on next page*

*Figure 3 continued*

morphological recoveries from Lis1 mutants crossed to a *Calb1-cre*;Ai14 mouse line (n = 8 total) – filled stars have confirmed calbindin expression and open stars are calbindin negative recordings. These cells are then run through the same clustering algorithm, and the associated LRI/ORI positions are plotted over the original clustering. Note filled and open stars fall in the upper right (complex) quadrant and the lower left (simple) quadrant, respectively. (C) Example morphological reconstructions, ranging from most complex (*left*) to simple (*right*). (D) Positional properties for predicted calbindin (complex, filled circles) and non-calbindin expressing (simple, open circles) principal cells. Note predicted calbindin expressing cells were superficial to non-calbindin predicted, and this trend was inverted for Lis1-MUT animals. Wt: complex 36.42 ± 8.5 µm, simple 50 ± 6.9 µm; Lis1-MUT: complex 128 ± 26.6 µm, simple 90.9 ± 12.7 µm, n = 13, 11, 8, 13, respectively. Depth is measured as it was for *Figure 2* from the front/superficial side of the PCL. (E) Group sorted measurements for distance along primary apical dendrite until first prominent bifurcation occurs. Wt: complex 163 ± 32.8 µm, simple 295.9 ± 41.4 µm; Lis1-MUT: complex 241.6 ± 55.8 µm, simple 368.9 ± 43.4 µm. Note complex cells tend to bifurcate sooner in both mutant and non-mutants. (F) Sholl interactions from Wt apical dendrites alone, of complex and simple sorted cells. (G) Likewise, for Lis1-MUT animals. (H) Overlay of the complex morphology sholl data from non-mutant and mutant experiments. Despite retaining a relatively complex population, complex Lis1-MUT principal cells have decreased apical dendritic branching that peaks closer to the soma. (I–K) Somatic images confirming calbindin expression from three recordings performed in Lis1 mutants bred to a *Calb1-cre*;Ai14 line. Morphologically these cells clustered with the complex group in (B) – filled stars, while calbindin negative recoveries (not shown here) are plotted as open stars in (B). Dashed circle diameter is 15 µm.

The online version of this article includes the following source data for figure 3:

**Source data 1.** LRI/ORI values from script, and resulting morphological group.

positioned complex cells in Lis1-MUT experiments). Grouping cells by the assigned shape cluster and plotting the associated PCL depth measurements (from the border between the first PCL and the radiatum) gives further support to the sorting results, as complex cells were located superficial to simple cells in normal type controls and scattered but generally deeper than simple cells in Lis1-MUT mice, in agreement with our calbindin staining experiment (*Figure 3D* PCL depth, Wt complex: 35 ± 8.0 µm, simple: 50.2 ± 6.3 µm, n = 13 and 11; Lis1-MUT, complex: 127 ± 23.4 µm, simple: 94.6 ± 12.3 µm, n = 8 and 13). Additionally, we observed that complex cells in both mutant and non-mutant animals tended to have their first prominent apical branch bifurcation points sooner than simple cells (*Figure 3E*). This suggests that the complex apical branch morphology can still be used to identify putative calbindin-expressing principal cells in Lis1-MUT mice. It should be noted that the clustering algorithm has no direct knowledge of somatic positioning, or what is determined to be the first primary apical bifurcation – yet these differences appear using the labels assigned in the clustering. More traditional morphological analyses such as sholl intersections fail to show clear differences between complex and simple cell types when they are pre-sorted by K-means label, highlighting the usefulness of analyzing branching patterns with this approach (*Figure 3F–H*, note that sholl intersections represent apical dendritic trees only and do not include basal dendrites). For additional confirmation that calbindin remains predictive of complex morphologies in Lis1-MUT animals, we crossed mice to a *Calb1-cre*:Ai14 reporter line and made recordings in these mutants and processed their morphological reconstructions through the algorithm (*Figure 3B*; open and filled stars). Out of eight successful recoveries, four principal cells with confirmed calbindin expression had LRI and ORI values in the upper right (complex) quadrant (*Figure 3I–K*). The remaining were calbindin negative and had relatively simple morphological values (*Figure 3B*, lower left). An additional three calbindin positive recordings are not included in the analysis, as their recoveries were split across multiple sections or incomplete, but these showed the hallmark of an early bifurcation point and dense early branching. These data support the notion that relatively complex and simple morphologies persist in the Lis1 mutant, and calbindin-expression remains predictive of the complex morphological group.

Readers should note, direct comparisons of sholl analysis from mutants and wild-type litter mates revealed a reduction in branch intersections in Lis1-MUT complex cells (*Figure 3H*). While wild-type complex cells typically have peak sholl intersections of 8–11 around 200 µm from the soma, Lis1-MUT complex cells have fewer peak intersections (~7), closer to the soma (~125 µm) (Wt n = 10 complex and 14 simple; Lis1-MUT n = 10 complex and 12 simple). While relatively speaking, the complex and simple subtypes persist in the Lis1-MUT mice, there has been an effect of the mutation, either direct or indirect, in stunting general morphological development. Clustering was performed separately for mutant and non-mutant data to reveal relative relationships between sub-types of cells in one genetic condition rather than an absolute comparison of all cell morphologies, allowing us to compare how related cell subtypes and their associated microcircuits develop under heterotopia.

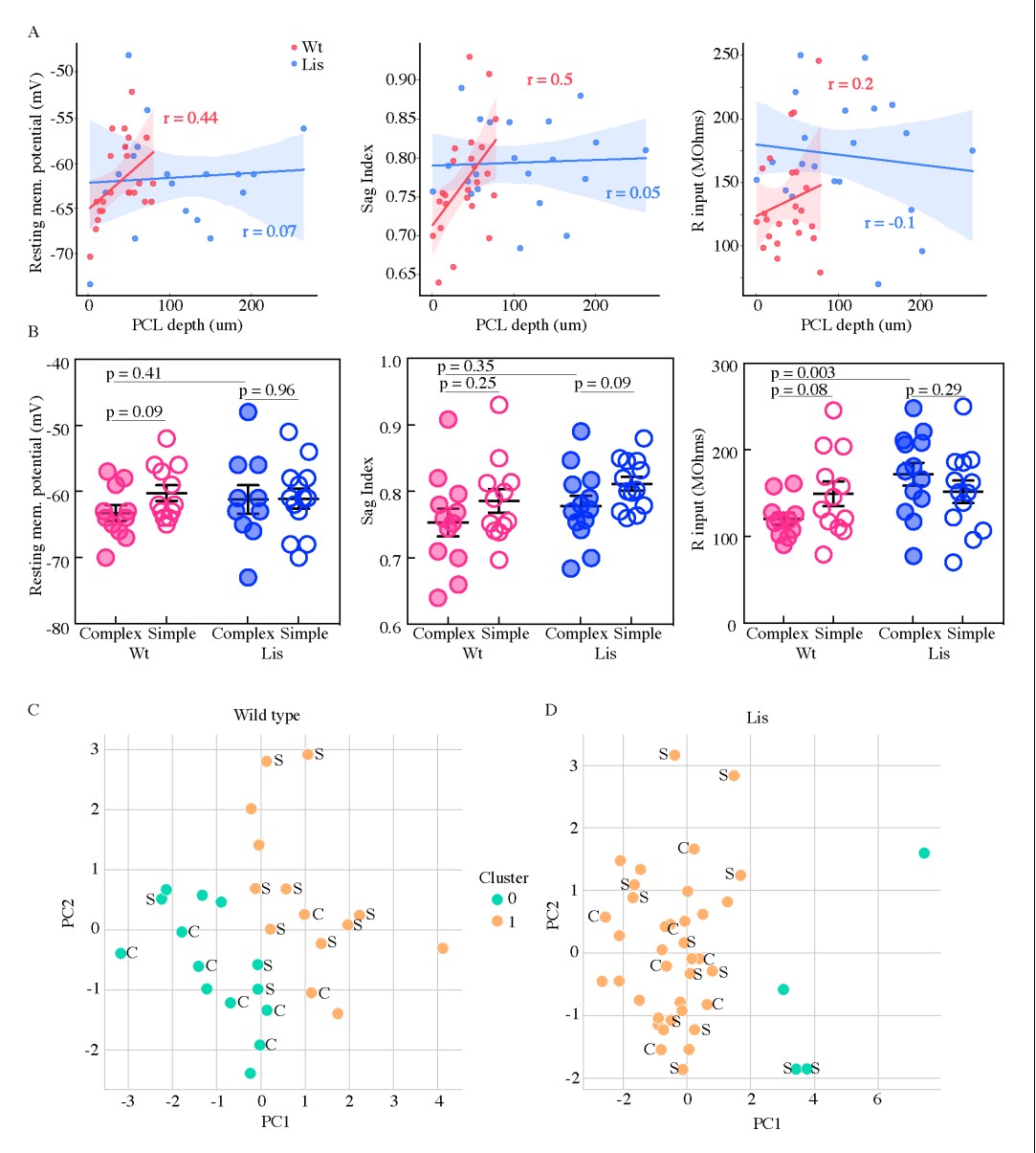

**Figure 4.** Physiological properties of calbindin positive and negative morphological clusters. (A) *Left*, somatic PCL depth correlations with cellular resting membrane potential for wild-type (red) and Lis1-MUT (blue) recordings. *Middle*, likewise, for sag index, where values closer to 1 correspond to less sag exhibited. *Right*, same for input resistance. (B) Same data as in (A), grouped by predicted calbindin expression. (C) Supervised K-means (n = 2) sorting wild types. A handful of electrophysiological properties alone are capable of reasonably accurate morphological subtype prediction (and therefore calbindin expression). C's and S's correspond to the data points associated morphological group, note that even mis-categorized points are near the midline. Of 8 morphologically complex cells, 6 are found in in physiological cluster 0, of 11 simple cells, 8 are found in physiological cluster 1. (D) Same as in (C) for recordings in Lis mutants. Physiological properties are less capable of predicting morphological cluster in Lis mutant mice. The online version of this article includes the following source data for figure 4:

**Source data 1.** Physiological properties by morphological type, position within CA1, and clustering data/results for ephys Kmeans.

## Lis1 mutant principal cells display disrupted physiological properties

From the whole-cell recordings that were used for morphological reconstructions in *Figure 3a* battery of intrinsic physiological properties were analyzed in two ways. Several of these properties are shown in *Figure 4*. Each property was plotted against the PCL depth of the soma (somatic depth from the radial side of the PCL) from which the recording was made (*Figure 4A*). The same data were also sorted into putative calbindin-positive and calbindin-negative cell types as predicted by

either complex or simple morphologies (*Figures 3B* and *4B*). Resting membrane potential displayed a pearson r value of 0.44 for correlation with position in wild-type litter mates, and a r-value of 0.07 in Lis1 mutant mice (Wt: n = 23, Lis1-MUT: n = 23). Sag index correlated with position at an r-value of 0.5 in wild-type mice and an r-value of 0.05 in Lis mutants (Wt: n = 24, Lis1-MUT: n = 26). Input resistance and depth in wild-type mice had a correlation value of r = 0.2, while in Lis mice r = −0.1 (Wt: n = 23, Lis1-MUT: n = 26).

In sorting recorded data by putative cell type, we noted that many of the positional differences observed in *Figure 4A* persisted or at least trended toward significant in wild-type littermates (complex cells are filled circles, open are simple; Resting membrane potential: Wt mean complex −63.3 ± 1.2 mV, simple −60.3 ± 1.2 mV, p=0.09 n=11 and 12; Sag index: mean complex 0.75 ± 0.02, simple 0.79 ± 0.02, p=0.25 n=12 and 12; Input resistance: complex 120.4 ± 6.8 MΩ, simple 149.3 ± 14.11 MΩ, p=0.08 n=11 and 12). Some of these differences in sub-types were still detectable in Lis1 mutants, but differences between principal cell sub-types for most properties seemed substantially diminished from those in normal mice (RMP: mean complex −61.2 ± 2.1 mV, simple −61.1 ± 1.5 mV, p=0.96 n=10 and 13; Sag index: mean complex 0.78 ± 0.02, simple 0.81 ± 0.01, p=0.09 n=13 and 13; R input: mean complex 171.9 ± 13.2 MΩ, simple 123.3 ± 13.02 MΩ, p=0.29 n=13 and 13). Note that not all physiological recordings had associated morphological recoveries. Additionally, in some recordings not all properties were measured, meaning some morphological groups vary in their summary N's.

We wondered if there were physiological subtypes of principal cells and how those subtypes might correspond to our previously identified morphological subtypes of complex and simple. Principal component analysis and subsequent K-means clustering was carried out on the physiological data (*Figure 4C and D*, resting membrane potential, sag index, input resistance, spike amplitude, adaptation ratio, firing frequency at 2x threshold, spike threshold, and after hyperpolarization amplitude were used for physiological clustering). Note that the clustering in this figure has no knowledge of morphological features nor LRI or ORI values from *Figure 3B* – we merely carry over the morphological labels afterward. Pre-clustering analysis with nbclust and elbow plots was performed, both approaches suggested n cluster = 2 was the optimal solution (data not shown). We then scored where morphologically identified cells fell in the physiological clusters. Out of eight morphologically complex cells, six were found in physiological cluster 0 and the remaining two in physiological cluster 1. Of eleven morphologically simple cells, eight were located in physiological cluster 1 and the remaining three in cluster 0, suggesting that these physiological clusters roughly correspond to the two morphological subtypes identified in *Figure 3* for wild-type littermates (*Figure 4C*). The same analysis in Lis1 mutants yielded uneven cluster counts, and no clear relationship between physiological cluster and morphological cluster (*Figure 4D*. The data indicate a loose relationship between morphological subtype and physiological subtype in wild-type animals that has been significantly perturbed under the Lis1 mutation - suggesting that physiological aspects of cellular identity may become smeared or lost under cellular heterotopia before or to a greater extent than morphological aspects. Put another way, cellular morphology is less predictive of intrinsic physiological properties under the Lis1 mutation.

## Basket cell – principal cell innervation biases are differentially affected in the Lis1 mutant hippocampus

Having gained insight into how the heterozygous *Pafah1b1* mutation impacts the development of principal cell properties of positioning, embryonic birthdate, morphology and intrinsic physiology, we next wondered how ectopic calbindin cells were integrated into the local synaptic network of CA1. Prior studies have suggested a preferential and complementary innervation bias among two types of local basket cells found in the CA1 subfield – parvalbumin-containing (PV) and a subset of cholecystokinin-containing (CCK) inhibitory interneurons. PV-expressing basket cells preferentially innervate deeply situated calbindin-negative principal cells, while CCK-expressing interneurons have a similar bias, but for superficial calbindin positive principal cells (*Lee et al., 2014*; *Valero et al., 2015*; *Valero and de la Prida, 2018*). We wondered if these innervation patterns were present in the Lis1 mutant despite ectopic cellular layering, which might shed light on how positioning and layering effect synaptic network development of brain structures.

To begin to assay this network feature in our Lis1 mutants, we first asked where these two types of basket interneuron somas were positioning in mutant mice. Immunohistochemical staining

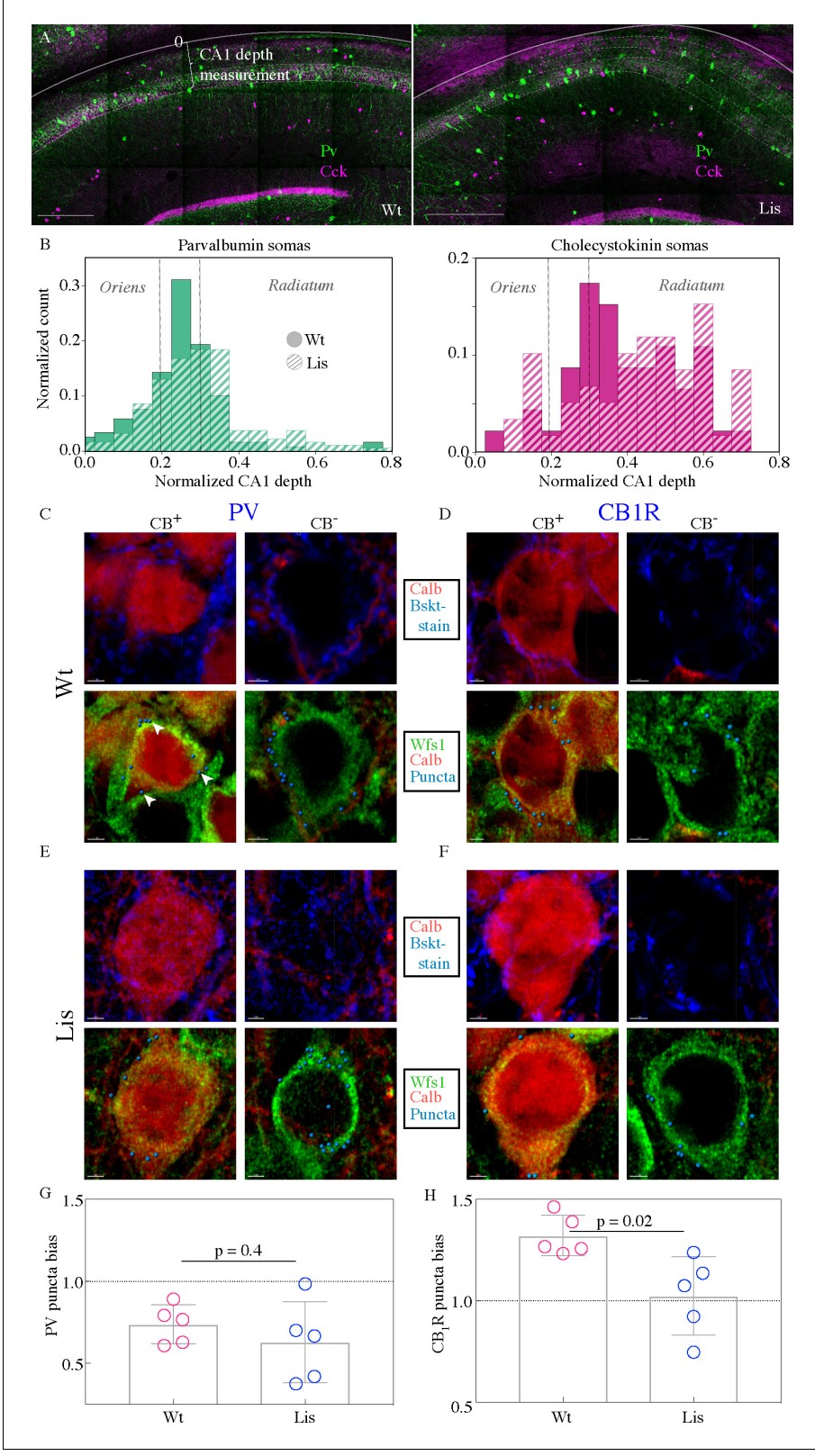

**Figure 5.** CCK-expressing basket cells have decreased innervation preference with ectopic calbindin-positive principal cells. (**A**) Low-magnification images showing the locations of parvalbumin and cholecystokinin-expressing interneurons in the CA1 hippocampus. Note the CA1 depth measurement from the back of the oriens – this measure is more appropriate for assessing somatic position within the larger CA1 structure, as opposed to PCL
*Figure 5 continued on next page*

*Figure 5 continued*

depth used elsewhere in this study. Scale bars are 350 µm. (B) Normalized histograms of basket cell soma depth measurements along the radial axis of CA1, both PV- (*left*) and CCK-containing (*right*) inhibitory interneuron somas show modest superficial shifts in Lis1-MUT mice. (C) High-magnification images of a staining experiment for the quantification of PV-containing inhibitory puncta from control littermate samples. *Left*, an example CB-expressing principal cell. *Right*, an example non-CB-expressing principal cell. The top row shows calbindin and parvalbumin staining, the bottom row shows the same cells with calbindin, Wfs1 staining which was used to draw the cell border, and the puncta derived from the parvalbumin staining shown above (arrows point to a few in the first panel) – these puncta are filtered for proximity to a postsynaptic gephyrin puncta (channel not shown). (D) Same as in (C), except the interneuron staining is for the cannabinoid receptor 1, highly expressed in the terminals of CCK-expressing interneurons. (E and F) Same as the corresponding above panels, but for samples from Lis1-MUT littermates. Scale bars are 2 µm. (G) PV puncta bias summary. PV puncta had a modest preference for non-calbindin expressing principal cells in both non-mutant and mutant slices. PV-calbindin preference: 0.74 ± 0.05 and 0.63 ± 0.11 innervation biases for wild-type and mutants, respectively, p=0.55, each point represents 12 cells from a slice, n = 3 pairs of littermates from three litters. (H) Same as in (E), but for experiments where the PV antibody was replaced by the CB1-R antibody. Non-mutant CCK baskets displayed a preference for calbindin-expressing principal cells that was lost in Lis1-MUT mice. CB1-R-calbindin preference: 1.32 ± 0.04, 1.02 ± 0.09 for wild-type and mutant respectively, p=0.02. Scale bars for C-F are 2 µm.

The online version of this article includes the following source data for figure 5:

**Source data 1.** Summary interneuron positions within CA1.

---

experiments were performed using antibodies against PV and CCK (*Figure 5A and B*). The somas of stained interneuron classes are plotted in binned and normalized histograms in *Figure 5B*, left and right for PV and CCK, respectively (filled bars for Wt dashed bars for Lis1-MUT). Vertical dotted lines show the approximate location of the wild type principal cell layer. Note that for this figure, somatic position is measured from the alveus/cortical border toward the s. radiatum across the entire radial depth of CA1, as opposed to how it is measured when examining principal cell layer depth, meaning 0 corresponds to the deepest position in this plot. This measure is more appropriate when looking at non-principal cells and overall hippocampal distributions (compare with *Figures 3F* and *4A*), as these interneurons often position on the edges of, or outside of the PCL. Our data indicate that both PV- and CCK-containing cell types have undergone superficial radial shifts, that is, the cell bodies have moved toward the s. radiatum. Notably, this is opposite the direction in which calbindin positive principal cells are shifted in Lis1 mutants (*Figures 1* and *2*). Overall, PV-containing somatic shifts appear less severe than CCK-containing shifts, but in both cases a few drastically shifted somas were observed (right tail of dashed histograms).

To begin to probe synaptic network development under heterotopia, we performed high-magnification immunohistological staining experiments with four simultaneously visualized channels (*Figure 5C-F*). This permitted the identification of inhibitory synapses on the somas of calbindin-positive and calbindin-negative principal cells (*Figure 5C*, *left* and *right* panels, respectively) in normal and Lis1 mutant littermates (5C vs E and 5D vs F, for PV and CB1R respectively). First, putative inhibitory boutons are automatically identified in the corresponding stain (Pv or CB1-R, top panels, blue staining). These putative pre-synaptically localized boutons are then filtered by proximity to a post-synaptic inhibitory synapse marker, gephyrin – yielding 'true' inhibitory puncta (synthetic spheres in bottom panels, gephyrin staining not shown). These puncta are then counted if they are within 0.2 µm or less of a principal cell soma – which are demarcated by the WFS1 antibody (green). Six calbindin positive and six calbindin negative principal cells in CA1 of mutants and non-mutant littermates are used for each image, yielding a single data point – that is to say, filtered inhibitory puncta are counted on somas of six calbindin positive and six calbindin negative principal cells. Six cells are used because sections and images taken are extremely thin to minimize Z-axis problems. We want to analyze ectopically positioned calbindin-expressing principal cells, and there are a limited number of these in any given image. The counts on calbindin-positive somas are divided by counts on calbindin-negative somas yielding a bias ratio (no. of Calb-positive PCs/no. of Calb-negative PCs). Numbers greater than one indicate a preference for calbindin-expressing principal cells. See Materials and methods for additional information.

PV-expressing basket cells preferentially innervated calbindin-negative principal cells in both mutant and wild-type mice (*Figure 5G*; PV-calbindin preferences: 0.74 ± 0.05, 0.63 ± 0.11 for Wt

and mutant, respectively, p=0.55, each point represents 12 cells from a slice, n = 3 pair of littermates from three litters). In experiments where the PV channel stain was replaced with a Cb1-R antibody, known to selectively stain presynaptic terminals of CCK-expressing basket cells, we noted a preferential innervation of calbindin-expressing post-synaptic targets in normal type that was absent from the Lis1 mutant mouse (*Figure 5H*; CB1-R-calbindin preferences: 1.32 ± 0.04, 1.02 ± 0.09 for Wt and Lis1-MUT, respectively, p=0.02). Which suggested that at least from an immunohistological level, CCK-expressing basket targeting onto ectopic calbindin-positive principal cells was disrupted.

## Monosynaptic CCK-mediated inhibition onto calbindin-positive principal cells is disrupted in CA1 of the Lis1 mutant

In order to better understand the role of CCK-expressing inhibitory cell networks in the face of pyramidal cell heterotopia and to further the observations shown in *Figure 5* at a functional level, whole-cell recordings were made from principal cells in CA1 in the presence of excitatory synaptic transmission blockers (APV 50 uM and DNQX 10 uM). Monosynaptic inhibitory events were evoked using a stimulation electrode placed locally in the PCL of CA1, and omega-conotoxin (1 µM) was applied to selectively inhibit vesicle release from CCK-expressing interneurons (*Figure 6*; *Heft and Jonas, 2005*). Example traces from four groups are shown in *Figure 6C*, from left to right, Wt complex, Wt simple, Lis1-MUT complex, Lis1-MUT simple. Baseline events are in black, and post wash-in data are in gray. In littermate controls, conotoxin reduced monosynaptically evoked IPSCs to 52.5 ± 3.9% of baseline amplitudes in complex cells, while events in simple cells were reduced to 75.6 ± 8.3% of baseline amplitudes, consistent with our observation that complex cells are preferentially targeted by CCK-containing interneurons (*Figure 6D* (*left*), p=0.03, n = 8 Wt and 8 Lis1-MUT cells). In Lis1 mutant mice this differential CCK-containing inhibitory input was not detected, as conotoxin reduced eIPSCs to 48.2 ± 16.4% of baseline and 60.2 ± 7.8%, for complex and simple cell subtypes respectively (*Figure 6D* (*right*), p=0.53 n=5 and 13).

We next repeated this experiment using an antagonist known to inhibit release from parvalbumin-expressing interneurons, omega-agatoxin IVA (250 nM). Example traces for the four subtypes before and after agatoxin application are shown in *Figure 6E* (wash-in data in gray). In control mice, agatoxin reduced monosynaptically evoked eIPSCs to 42.01 ± 6.2% of baseline in complex cells, events in simple cells were reduced to 9.5 ± 0.7% of baseline amplitudes, signifying that events in simple cells were more dependent on PV-expressing basket cell input (*Figure 6F* (*left*), p=0.003, n = 6 complex and 4 simple cells). In Lis1 mutant mice agatoxin reduced eIPSCs to 49.6 ± 5.8% of baseline and 14.2 ± 3%, for complex and simple cell subtypes, respectively (*Figure 6F* (*right*), n = 6 and 7, p=0.0002). In comparing both interneuron networks between genotypes, data suggest that CCK innervations are more perturbed that PV in Lis1 mutants. In both monosynaptic wash-in experiments representative traces come from single examples, but readers should attend to the spread of the points particularly in the mutant data sets. We believe this variance to largely stem from differences in severity of heterotopic banding between animals or slices. For example, not all Lis1 mutant slices display severe heterotopic banding, but instead a scattering of cells at various points of CA1. Additionally, it is difficult to assess the degree of banding prior to making whole-cell recordings. Hence, some calbindin-positive complex cells in mutants may in fact come from the superficial heterotopic band, while others (the ones we select for when doing immunohistochemical analysis particularly in *Figure 5*) are located ectopically in the deeply positioned heterotopic band. In most cases, we see the greatest deficits in the wash-in experiments, in these heterotopic principal cells, and less severe deficits in the more normally positioned complex cells in the Lis1 mutants. While this complicates analysis by increasing the putative groups or obscuring relationships when grouping all cells by shape rather than position and shape, the data seem to hint there is indeed something about the disrupted layering itself, or the heterotopic positioning that is contributing to the failures in microcircuit formation observed here.

Having probed monosynaptic inhibitory circuitry onto putative calbindin-positive and -negative cells, we next examined feedforward disynaptic inhibition onto CA1 principal cells in normal and Lis1 mutant mice. Superficial cells have been previously shown to exhibit a comparatively higher level of excitatory drive during feedforward circuit activation (large EPSCs per unit of IPSC, *Valero et al., 2015*). Cells were voltage clamped at −70 mV and +10 mV to measure the Schaffer collateral-mediated monosynaptic excitatory and disynaptic inhibitory drive (*Figure 6G*). Excitatory transmission was subsequently blocked (APV 50 µM and DNQX 20 µM), to allow the subsequent isolation of the

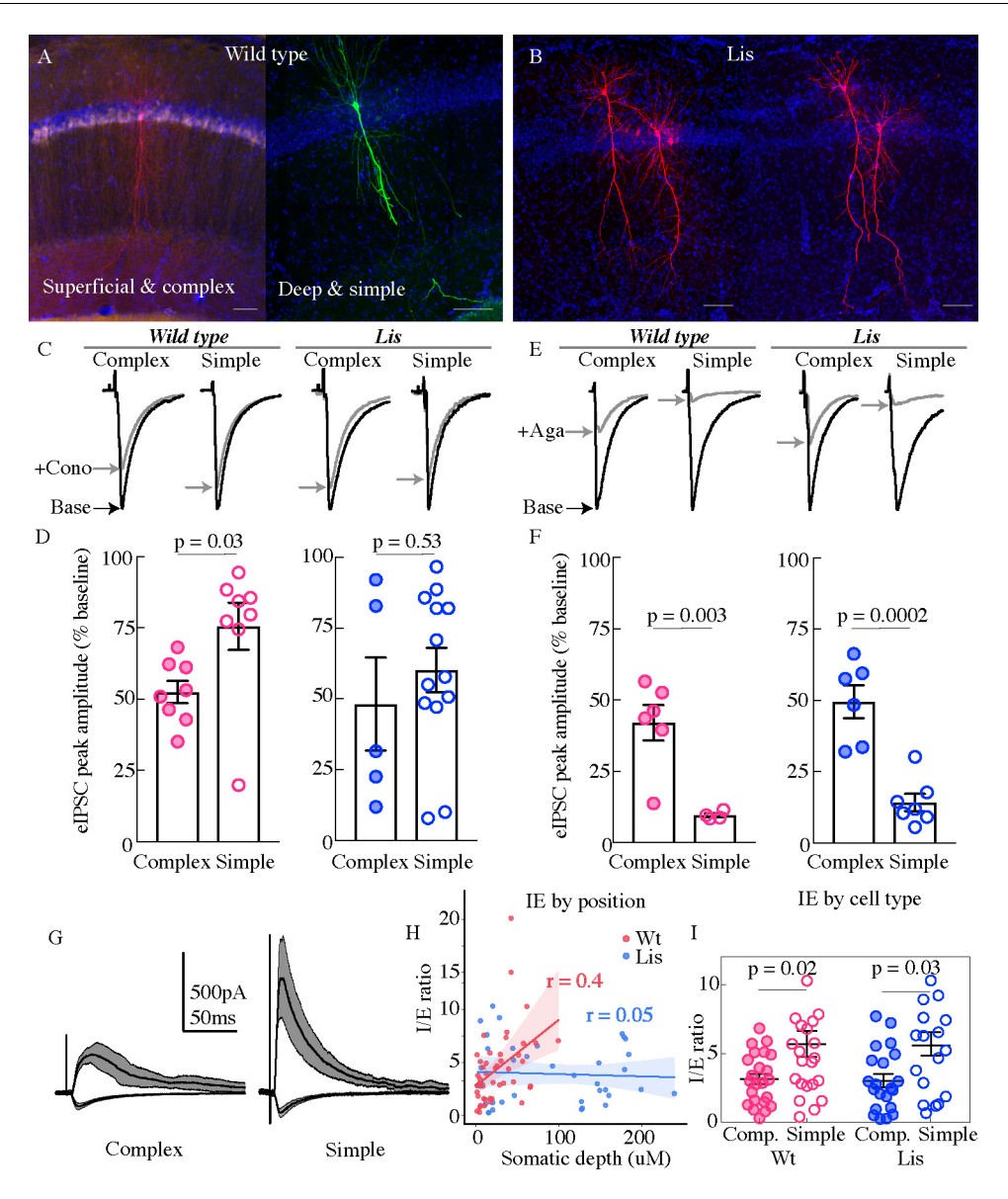

**Figure 6.** Physiological assays of network function within CA1. (**A** and **B**) Cell recoveries from normal type and Lis1 mutant experiments. Scale bars are 85 μm. (**C**) Normalized example traces from pre- and post-wash in (dashed) of omega-conotoxin (1 μM), *from left to right*, a normal-type complex and simple recordings, followed by Lis mutant complex and simple examples. Stimulation for monosynaptic experiments was delivered locally in the CA1 PCL. (**D**) Quantification of the percent reduction in the evoked IPSC 10–12 mins after drug application. Wt: complex 52.5 ± 3.9%, simple 75.6 ± 8.3%; Lis1-MUT: complex 48.2 ± 16.4%; simple 60.2 ± 7.8%, n = 8, 8, 13, 5, respectively. (**E**) Example traces as in (**C**) but for omega-agatoxin experiments (250 nM). (**F**) As in (**D**) but for agatoxin. Wt: complex 42.0 ± 6.2%, simple 9.5 ± 0.7%; Lis1-MUT: complex 49.6 ± 5.8%; simple 14.2 ± 3.0%, n = 6, 4, 6, 7, respectively. (**G**) Example traces for monosynaptic EPSCs (excitatory, inward current), and disynaptic feedforward IPSCs (inhibitory, outward current) evoked by stimulation of Schaffer collaterals, from a simple and complex recovered cell morphology in normal type. (**H**) IPSC amplitude/EPSC amplitude plotted by somatic PCL depth. (**I**) Same data as in (**H**) sorted by cell sub-type. Wt: complex 3.15 ± 0.39, simple 5.70 ± 0.95; Lis1-MUT: complex 3.02 ± 0.49%; simple 5.03 ± 0.76, n = 23, 23, 21, 17, respectively.

The online version of this article includes the following source data for figure 6:

**Source data 1.** Monosynaptic wash-in event reduction percentages; disynaptic data summary and IE ratios with assigned morphological group.

disynaptic feedforward inhibitory drive from the total inhibitory component. Inhibition:excitation (IE) ratios were positively correlated with somatic depth in the PCL for wild-type littermates, but not Lis1 mutants (*Figure 6H*; Wt r = 0.4, Lis1-MUT r = 0.05). When recorded cells were sorted by complex and simple morphologies complex cells had lower IE ratios in both wild-type and Lis 1 mutant mice (*Figure 6I*, Wt complex 3.15 ± 0.39, simple 5.7 ± 0.95, p=0.02 n=23 complex and 23 simple; Lis1-MUT complex 3.02 ± 0.49, simple 5.03 ± 0.76, p=0.03 n=21 complex and 17 simple cells). While their resulting ratios were predictive of sub-type, neither EPSC or IPSCs alone were significantly associated with depth or cell subtype (data not shown). EPSCs displayed depth correlations of r = 0.16 and r = 0.07 for wild-type and Lis1-MUT experiments, respectively. Neither excitatory nor inhibitory events differed significantly between principal cell shapes. IPSCs had a somatic depth correlation value of 0.2 wild-type littermates and 0.01 for mutants.

## Lis1-MUT mice display robust extracellular oscillations but are less synchronous across heterotopic bands

Using extracellular oscillations measured in vitro we next sought to assay alterations in network level function resulting from the cellular heterotopia present in our Lis1 mutants. Both wild-type and Lis1 mutant slices were capable of producing robust gamma oscillatory activity (ranging from 18 to 50 Hz), in response to application of 20 µM carbachol (*Figure 7*; *Buhl et al., 1998*; *Fellous and Sejnowski, 2000*; *Fisahn et al., 1998*). Slices from Lis1 mutants produced slightly higher frequency gamma oscillations than non-mutants (Wt 24.9 ± 1.7 Hz, Lis1-MUT 31 ± 1.1 Hz, p=0.005 n=20 and 14, respectively) (*Figure 7B–D*). Subsequent addition of the synthetic CB1R agonist, WIN-55,212–2 (WIN) (2 µM), did not alter the peak frequency of the oscillations in normal type nor mutant recordings (*Figure 7D*) but caused a significant decrease in peak power in normal type recordings (*Figure 7E*), but not in Lis1-MUT mice suggesting that CCK-networks in mutants are less involved in gamma oscillation generation than in wild-type littermates (Wt +WIN 0.93 ± 0.03 vs CCh alone p=0.03, Lis1-MUT +WIN 1.02 ± 0.04 vs CCh alone p=0.69; n = 20 and 14 non-mutant and mutant, respectively).

In an additional series of experiments, a second electrode was placed in the same radial axis as the first approximately 150 µm deeper, so that in normal type slices one electrode targeted the radiatum side of the PCL while the other targeted the oriens side (*Figure 8A*). In the Lis1-MUT slices, electrodes were placed in different heterotopic bands but still in the same radial axis. This allowed for analysis of the correlation and synchronicity of oscillations across the normal and heterotopic layers of CA1 (*Figure 8*). Electrode location was preserved in analysis such than comparisons are always made in a deep vs superficial manner. Examples of simultaneous one second recordings are shown for the oriens (*top*) and radiatum (*bottom*) side electrodes in *Figure 8B* (Wt on *left*, Lis1-MUT on *right*). Dashed vertical lines show peak alignment for each example. Associated cross-correlation plots between these electrodes are displayed in *Figure 8C* (Wt *left*, Lis1-MUT *right*); note the +0.7 ms peak in offset in the wild-type experiment, and −2.7 ms peak offset in the Lis1 example. Wild-type and Lis1 mutant slices were capable of producing correlated oscillatory activity (*Figure 8D*; Wt 394.6 ± 80.0, Lis1-MUT 394.2 ± 60.8, p=0.99 n=20 and 14). However, examining the time-shifts obtained from cross correlation analysis (how far one signal is peak shifted from another in time) we noted that Lis1-MUT mice displayed significantly less temporally correlated oscillations between the two electrodes (*Figure 8E*; Wt: +1.01 ± 0.8 ms, Lis1-MUT: −1.8 ± 0.79, p=0.02 n=20 and 14) suggesting that while both heterotopic bands participate in the ongoing oscillation, their separation in anatomical space or deficits in basket cell network connectivity erodes the correlated activity between the bands. Application of the CB1R agonist WIN-55 produced modest decreases in wild-type cross-correlation values but not in the Lis1 mutants (Wt: + WIN 333.9 ± 71.9, vs baseline p=0.04 n=20; Lis1-MUT: + WIN 427.2 ± 84.13, vs baseline p=0.43 n=14) suggesting a diminished role for CCK-containing interneuron networks in the mutant mouse. WIN-55 application did not have a significant impact on the time-shift between deep and superficial channels in either genetic background (Wt: + WIN 0.68 ± 0.52 ms, vs baseline p=0.62, Lis1-MUT: + WIN −0.41 ± 1.23 ms, vs baseline p=0.21).

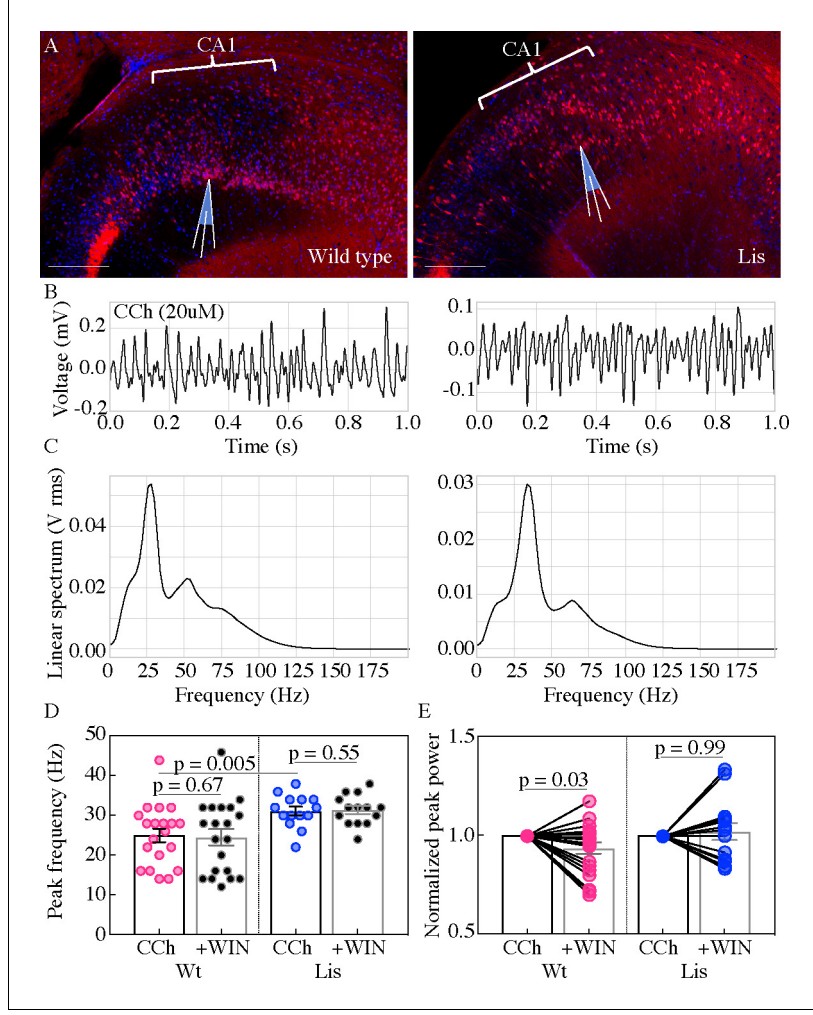

**Figure 7.** Lis mutants display robust carbachol-induced oscillations. (**A**) Normal type (*left*) and mutant (*right*) images from ventral hippocampus in *Calb1-cre:*Ai14 mice. Note the second layer of deeply positioned calbindin expressing principal cells in the Lis mutant. Scale bars are 200 µm. (**B**) One second of data during carbachol induced activity from radiatum side electrodes in normal type and mutant recordings, respectively. (**C**) Power spectra computed for each of the above example recordings. (**D**) Summary peak frequency data for wild-type and Lis mutant experiments, in carbochol alone, and with addition of WIN-55 (Cb1-R agonist, 2 um). Wt CCh 24.88 ± 1.7 Hz, +WIN 24.4 ± 2 Hz, Lis1-MUT CCh 31 ± 1.1 Hz, +WIN 31.3 ± 1 Hz. (**E**) Summary data as in (**D**) but for normalized Vrms power at the peak frequency. Wt +WIN 0.93 ± 0.03 vs CCh alone p=0.03, Lis1-MUT +WIN 1.02 ± 0.04 vs CCh alone p 0.69; n = 20 and 14 wild-type and mutant respectively. Pre-vs-post wash p values represent paired t tests.

The online version of this article includes the following source data for figure 7:

**Source data 1.** Peak frequency and peak power summary data., note we normalize power data with-in exp.

## Discussion

Cellular heterotopias arising from various genetic and environmental factors carry with them a poor prognosis for the affected individual, including severe mental disability, increased seizure risk, and shortened life span (*DE WIT et al., 2011*). The degree to which these effects are a direct result of the heterotopia itself (a lack of layers) or related to the role of the mutated genes in other processes remains unclear. That is to say, it is unknown to what extent any of the disease phenotypes associated with Lissencephaly are the result of disrupted layering and cellular misposition during embryonic development. However, by making relative comparisons between cell subtypes and their integration into the local circuitry separately in wild-type and mutant animals, we are able to garner

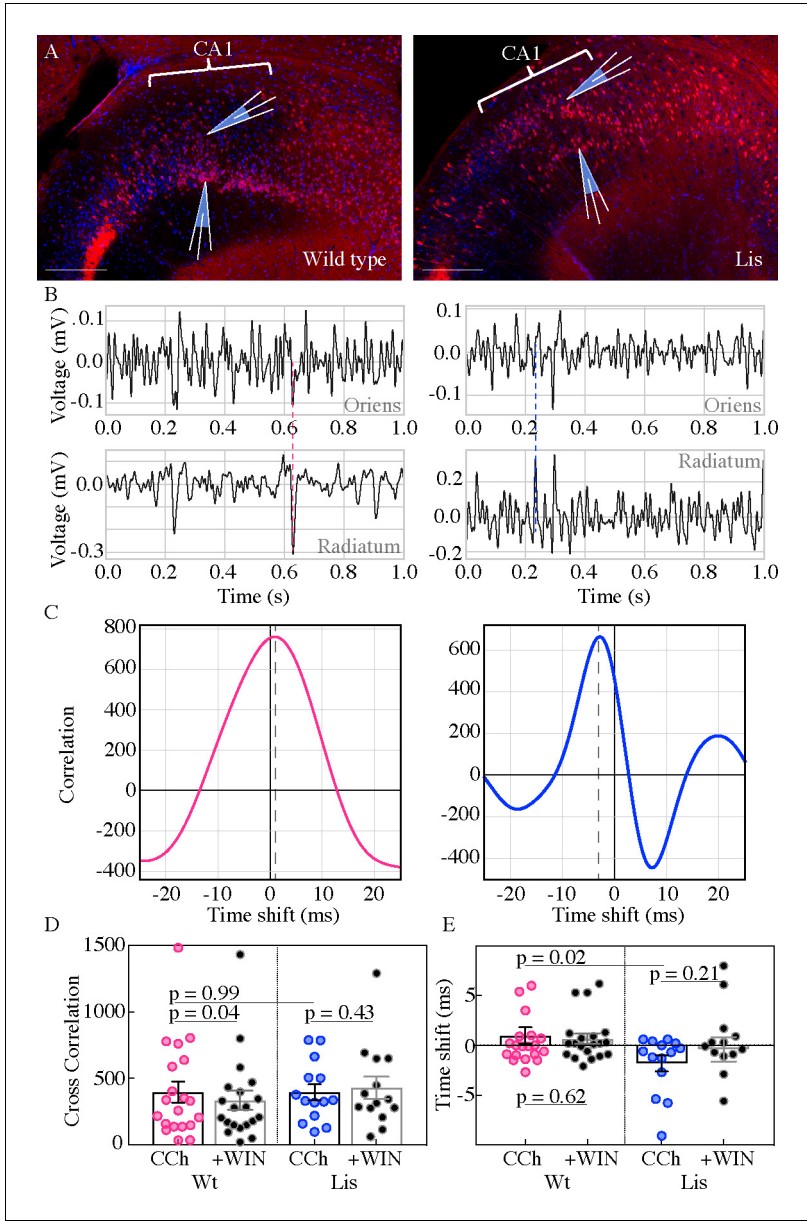

**Figure 8.** Carbachol oscillations in Lis1 mutants are less synchronous across CA1 heterotopias. (**A**) Normal type (*left*) and mutant (*right*) images from ventral hippocampus showing the positioning of dual electrode recordings, one from the s. radiatum and a second s. oriens side electrode in the same radial plane. Scale bars are 200 μm. (**B**) One second of simultaneous recordings from the deep (top) and superficial (bottom) electrodes, for wild-type (*left*) and Lis mutant (*right*) example experiments. Dashed lines highlight peak alignment between electrodes – note the blue line intersecting near a trough in the top trace, and a peak in the bottom. (**C**) Cross correlation plots for the example experiments shown in (**B**). Correlation values are arbitrary units. (**D**) Summary data for non-mutant and Lis1-MUT experiments in carbachol and after WIN-55 wash-in. Wt CCh 394.6 ± 80, +WIN 333.9 ± 72, Lis1-MUT CCh 394.2 ± 60.8, +WIN 427.2 ± 84.1. (**E**) Summary for the millisecond timing of peak correlation shifts shown in (**D**). Wt CCh 1 ± 0.8 ms, +WIN 0.68 ± 0.5 ms, Lis1-MUT CCh −1.8 ± 0.8 ms, +WIN −0.4 ± 1.23 ms; n = 20 and 14 wild-type and mutant respectively. Pre-vs-post wash p values represent paired t tests.

The online version of this article includes the following source data for figure 8:

**Source data 1.** Cross correlation and temporal shift data.

some insight into cellular maturation, subtype identity development, and susceptibility of key circuit motifs to a loss in layering.

In the present work, we first investigate the heterotopic banding observed in area CA1 of the Lis1 mutant mouse in order to determine if there is a pattern to the splitting of these excitatory cell populations – that might reflect naturally 'embedded' layers in the wild type CA1. To this end, we demonstrate that calbindin-expressing principal cells are preferentially affected by cellular heterotopia in CA1, where they are proportionately relegated to the deeper cellular layer – opposite of their normal superficial positioning in the PCL (*Figure 1*; *Slomianka et al., 2011*). After confirming that these cells are the same embryonically derived population (*Figure 2*), namely late-born calbindin expressing, we asked to what degree their intrinsic development reflected the differences between calbindin-positive and calbindin-negative PC subtypes in wild-type animals, and if relative differences between the two population were preserved (*Figures 3* and *4*). While there was an effect of stunted arborization in comparison to normal type calbindin cells, Lis1 calbindin cells retained their complex morphology relative to with-in animal non-calbindin expressing principal cells. Intrinsic physiological properties appear more disrupted in Lis1 calbindin-expressing principal cells; however, several properties showed greater differences or trended toward significant differences when separated by putative calbindin expression, as opposed to somatic positioning – suggesting again that subtype was a stronger influence than layering in the determination of these properties. It is unclear if the intrinsic physiological differences between calbindin positive PCs in normal and Lis1 mutants reflected other roles of the *Pafah1b1* protein directly, compensatory changes of ectopic cells, or are the result of cellular development in an ectopic position – the first two seem more likely given findings from other mis-lamination models (*Salinger et al., 2003*; *Wagener et al., 2016*; *Caviness and Rakic, 1978*), although insufficient circuit integration and activity is known to alter cellular development in cortex (*De Marco García et al., 2011*).

We next turned our attention to the integration of these ectopic calbindin-expressing principal cells into the CA1 basket cell network. Staining experiments suggest that CCK expressing basket cell synapses were specifically altered to a greater extent than PV networks onto ectopic calbindin principal cell targets (*Figure 5*). This finding was confirmed by monosynaptic inhibition experiments, which showed reduced sensitivity of ectopic calbindin-expressing principal cells to omega-conotoxin, which selectively impedes CCK cells –suggesting these connections are sparse or otherwise underdeveloped (*Figure 6*, left). Conversely, PV cell networks seemed substantially more resilient, which is not so surprising given that these cells occupy deeper positions within CA1, and their preferred synaptic targets are not substantially mispositioned under the cellular heterotopia present in Lis1 (*Figure 5A and B*; *Lee et al., 2014*). Interestingly, we observed greater spread in the mutant monosynaptic inhibition data that the wild-type counterpart. As briefly mentioned in the associated text, we suspect the spread stems from differences in the degree of heterotopic banding in any particular animal or slice. Data points showing greater inhibition deficits were often the cells whose soma were most ectopically located (calbindin cells far in the deeper heterotopic band). Notably, in *Figure 5* we select for these cells in our immunohistochemical bouton analysis, but this is much harder to do for whole-cell recordings – hence grouping the data by shape alone means some cells will be complex and calbindin expressing, but somewhat normally located. Others will be ectopically located complex cells – the population more likely to suffer from the developmental deficit. While this makes the data harder to gather, group, and analyze, this would suggest that some of the network deficits described here are not a de facto consequence of the Lis1 mutation – instead being tied to local CA1 architecture or loss of layering under heterotopia.

Disynaptic inhibition experiments also support the notion of PV networks being more robust under cellular heterotopia (*Figure 6*, bottom). Feed-forward inhibition is much stronger onto PV baskets than their CCK expressing counterparts, making this largely a test of PV network connectivity (*Glickfeld and Scanziani, 2006*). Additionally, depolarization to +10 mV (as done in these experiment) drives depolarization-induced suppression of inhibition in CCK-basket cells, largely removing them from this assay (*Freund and Katona, 2007*; *Lee et al., 2010*; *Neu et al., 2007*). In sorting these experiments by principal cell sub-type, we observed that ectopic calbindin-expressing principal cells retained their relatively high excitability (low I/E ratios), suggesting that parvalbumin cells did not start to inappropriately target deeply positioned, ectopic calbindin PCs.

Groups working in a related model of cellular heterotopia, the Reeler mouse which has severely disorganized cortical and hippocampal principal cell layering, previously reported that excitatory

and inhibitory cells are produced in approximately the correct proportions, that ectopic cells retain expression of their correct markers, morphology of cell types is generally conserved, and their intrinsic physiological properties are largely unperturbed on a network level (*Wagener et al., 2016*; *Boyle et al., 2011*; *Caviness and Sidman, 1973*; *Guy and Staiger, 2017*; *Guy et al., 2017*). Despite differing genetic causes, the present study supports these findings that brain development is surprisingly robust despite mis-lamination. An interesting caveat, however, is that in the present work and other studies of cellular heterotopias, morphological development and orientation of principal cell dendrites appear stunted and meandering (*Figure 3*; *Guy et al., 2015*; *Stanfield and Cowan, 1979*). In the Reeler mouse synaptic network development was also remarkably intact, as thalamocortical and intracortical connectivity, cellular tuning properties to stimuli, and even animal behavior seem only minorly altered if at all (*Salinger et al., 2003*; *Wagener et al., 2010*; *Wagener et al., 2016*; *Guy et al., 2015*). From a broad perspective, this is in agreement with the present work in the Lis1 hippocampus, as feed-forward properties onto PC subtypes retain their relative excitabilities, and Lis1 slices retain their ability to generate gamma oscillations (*Figure 7*).

Interestingly, we observed higher peak oscillation frequency in Lis1 mutant experiments than normal type (*Figure 7D*). One possible interpretation of this result is that CCK-expressing interneuron networks tend to generate lower frequency gamma, and when disrupted in Lis1 mutants, networks become more dependent on alternative faster oscillation mechanisms such as greater reliance on parvalbumin cell networks. These results may reflect biological differences in hyperexcitability that predispose these mice and human patients to seizures – further study is required to determine if more heavily banded hippocampal PCL regions in mutant animals have a greater propensity to act as seizure foci. In the power domain, measurements are sensitive to differences in electrode placement between experiments, as this cannot be ruled out particularly as the cell layer positioning is unruly in Lis1-MUT mice; power data from these recordings was normalized and only compared within experiment to wash-in values (*Figure 7E*). Non-mutant slices showed power decreases in the presence of the cannabinoid receptor agonist WIN-55, while Lis1 mutant slices were non-responsive to this compound. These data add to our immunohistochemistry and monosynaptic physiology experiments in suggesting deficits in the CCK-basket cell networks of CA1 under heterotopia as Lis1 slices are largely not affected by WIN-55 application.

Comparing recordings from two electrodes in *Figure 8* revealed that cross correlation values were relatively similar between wild type and mutant mice, but time-shifts or synchronicity between channels were significantly different (*Figure 8E*). It seems likely that timing differences in gamma-oscillations arise from the physical separation of current sinks and sources under Lis1-MUT heterotopia, and not as a result of the CCK-innervation deficit described above, as these measures were largely unchanged by WIN-55 application in normal-type mice, however, that possibility cannot be ruled out (*Hájos et al., 2000*; *Soltesz and Deschênes, 1993*). It is worth noting that the time-shifts under baseline conditions in the mutants are opposite in direction than that of non-mutants. In that respect, they roughly mirror the physical inversion of PCL lamina under Lis1-MUT cellular heterotopia.

Collectively, these findings bolster the notion that layers are in large part an epiphenomenon of neurogenesis, as has been hypothesized previously. Importantly, layer terminology has a correlated genetic component in wild-type mice as it is likely to capture a related embryonic pool of neurons. Therefore, when traditional studies refer to cellular layer, they are using it as a proxy for cellular genetic subtype, which is no longer the case in heterotopias (*Guy and Staiger, 2017*; *Caviness and Rakic, 1978*; *Guy et al., 2015*). In agreement with this line of reasoning, decades of work on synapse development are increasingly bolstering the 'hand-shake hypothesis' – where in molecular cues present on the surface of both putative synaptic partners confirm or reject synapse formation to aid in the establishment of appropriate and canonical circuitry over several scales of axon pathfinding (*Harris and Shepherd, 2015*; *Margeta and Shen, 2010*; *Blakemore and Molnár, 1990*; *Molnár et al., 2012*). The degree to which these genetic network wiring mechanisms are modified in an activity-dependent fashion afterword remains an area of active study (*Sur and Rubenstein, 2005*; *De Marco García et al., 2011*; *Che et al., 2018*). Importantly, the present study does identify a crucial network motif, CCK-interneuron targeting of calbindin positive principal cells, that is specifically disrupted in ectopic calbindin PCs in the Lis1-MUT mouse. Further work will be needed to determine if this is a genetically specified connection preference for calbindin-expressing principal cells, and why it might exhibit positional dependence.

It might not be so surprising to find specific defects in CCK-expressing synaptic connections as opposed to PV circuitry. CCK and PV expressing interneurons arise from different progenitor pools, in the caudal ganglionic eminence (CGE) and medial ganglionic eminence (MGE), respectively (*Butt et al., 2005*; *Fishell, 2007*). Additionally, CGE interneurons are developmentally lagged relative to MGE pools, as MGE cells are born first (*Tricoire et al., 2011*). Notably, later born basket cell populations (CCK basket cells), appear to be biased towards innervation of late born principal cell populations (superficial, calbindin expressing) in non-mutant animals. In fact, prior work has demonstrated that basket CGE derived populations wait until the first post-natal week to form synapses on principal cell somas in the PCL (*Morozov and Freund, 2003*). This network motif may represent a lopsided obstacle in the establishment of CA1 circuitry, as few if any of their putative synaptic targets remain on the radiatum adjacent side of the PCL under this form of cellular heterotopia (*Armstrong and Soltesz, 2012*). As CCK cell somas reside largely on the border between the PCL and the radiatum, in the Lis1 hippocampus these basket cells are tasked with sending axons through the denser superficial PCL and passing through the inter-PCL space before finding their appropriate synaptic targets in the deeper heterotopic band. It remains to be seen whether this CCK specific defect is generalized to area CA1 in other cellular heterotopias, or Lis1 specific, but it may suggest natural limits to the handshake hypothesis – after all if you are never introduced, you cannot shake hands.

## Materials and methods

### Animal care and breeding

All experiments were conducted in accordance with animal protocols approved by the National Institutes of Health. *Pafah1b1*$^{+/Fl}$ male mice (provided by the laboratory of Anthony Wynshaw-Boris, Case Western Reserve University) were crossed with *Sox2-cre* females (provided by National Human Genome Research Institute transgenic core, *Tg(Sox2-Cre)1Amc/J*). *Sox2-cre* females display cre-recombinase activity in gamete tissues, allowing us to genotype and select non-conditional Lis1-MUT mice without the *cre* gene in one generation. To identify mutant offspring, we designed a new forward primer (Recombined forward: AGTGCTGGGACAGAAACTCC, Reverse: CCTCTACCAC TAAAGCTTGTTC) from the previously published genomic sequences. These mice were bred to wild-type C57BL/6J mice (Jackson Labs stock no. 000664) and used for experiments. Both male and female *Pafah1b1*$^{+/-}$ mice were used for recording and immunohistochemical experiments. Female *Neurog2-Cre* (provided by the laboratory of Rosa Cossart, INSERM Marseille, France) mice were used for cell birth-dating experiments after being crossed to a cre-dependent reporter line (R26R, Jackson Labs stock no. 32037) – which contain an EGFP reporter with a *loxP* flanked stop cassette. *Calb1-cre* mice were obtained from Jackson laboratories (stock no. 028532) and bred to Ai14 animals from also from Jackson (stock no. 007914).

### Cellular birth-dating

Timed pregnancies were established between *Pafah1b1*$^{+/-}$ males and tamoxifen inducible *Neurog2-CreER:Rosa26* females. Tamoxifen administration in these pregnant mice induces cre-recombination and subsequent eGFP expression in newly born neurons of developing mouse pups. Pregnant mothers were gavaged with tamoxifen (Sigma no. T5648) in corn oil (200–250 µL, 20 mg/mL) at various embryonic time points spanning days E12-17. Pups were genotyped and grown to P27-32 before perfusion and brain fixation in 4% paraformaldehyde in 0.1 M phosphate buffer for 2–4 hr at room temperature or 12 hr at 4°C. Brains were washed, transferred to 30% sucrose in 1x phosphate buffered saline and stored at 4°C. Sections (50–100 µm) were cut on a frozen microtome and stained for calbindin protein (described below). Coronal hippocampal sections were confocally imaged under 20x magnification on a Zeiss confocal microscope, tiled, stitched in the Zen Black software package and post-hoc analyzed for colocalization of calbindin staining and eGFP expression using the Imaris analysis package (Imaris 9.3.1, Bitplane).

### Immunohistochemistry

Standard staining procedures were used for most of the experiments and have been described previously (*Chittajallu et al., 2013*) but briefly, deeply anesthetized mice were transcardially perfused

with 50 mL of 4% paraformaldehyde (PFA) in 0.1 M phosphate buffer (pH 7.6). Brains were post-fixed overnight at 4°C, then cryopreserved in 30% sucrose solution. Coronal sections were cut (50 µm) on a frozen microtome. Prior to staining sections are washed in phosphate buffered saline (PBS), blocked and permeabilized with 0.5% triton X-100, 10% goat serum in PBS for 2 hr at room temperature while shaking. Primary antibodies are applied overnight at 4°C shaking at the appropriate dilution with PBS containing 1% goat serum and 0.5% triton X-100. The following day sections are washed, and a secondary antibody is applied for 1 hr at room temperature while shaking at a dilution of 1:1000. For most experiments, a final DAPI staining was also used to show lamina of the hippocampus. Sections are then mounted and cover slipped with Mowiol. Primary antibodies: Calbindin (Millipore polyclonal rabbit, stock no. AB1778, 1:1000; or Swant monoclonal mouse 1:1000, stock no. 300); CCK (Frontier Institutes rabbit, stock no. CCK-pro-Rb-Af350, 1:1000).

For quantification of inhibitory puncta the procedure was similar with a few adjustments. Coronal sections (50 µm) of dorsal hippocampus were cut, blocked with 10% donkey serum in 0.5% Triton X at room temperature for 2–4 hr. Primary antibodies were applied in phosphate buffered saline with 1% donkey serum and 0.05% triton X-100 at 4°C for 48 hr. Secondary antibodies were left at room temperature for 1–2 hr, before washing and mounting. Primary antibodies: Gephyrin-mouse (Synaptic Systems, CAT no. 147021, 1:1000), Wolfram syndrome 1 (Wfs1)-rabbit (Protein Tech, CAT no. 1558–1-AP, 1:5000), cannabinoid1-receptor (CB1-R)-guinea pig (Frontier Institutes, CAT no. CB1-GF-Af530, 1:5000), parvalbumin (PV)-goat (Swant, CAT no. PVG 214, 1:5000). Calbindin was visualized by using pups from crosses between Lis mutants and *Calb1-cre*:Ai14 mice. Anti-donkey secondaries: Jackson Immuno Reseach laboratories Inc, AF 405 mouse (715-476-150), AF 488 rabbit (711-545-152), and AF 633 (706-605-148) guinea pig or goat (705-605-147) for visualization of CB1-R- and PV-positive baskets respectively (all 1:500). Images were captured on a Zeiss 880 confocal under 63x magnification using Zen Airyscan image processing. Between 25 and 30 Z-axis images were collected at Z-steps of 0.159 µm. Analysis was performed on a Max-IP from the first seven of these steps, accounting for 1.1 µm of tissue thereby minimizing Z-axis problems.

Images were quantified in Imaris 9.3.1 software. Twelve principal cells were selected using the Wfs1 staining – half of which were calbindin positive, and cell somas were traced. Gephyrin puncta (with an approximated size of ~0.25 µm) were automatically detected in the image and excluded if not within 1 µm of a cell soma. In parallel, inhibitory boutons were automatically detected from a pre-synaptic basket cell marker (parvalbumin in one set of experiments, CB1-R in the other). Inhibitory puncta were filtered for proximity to the post-synaptic gephyrin puncta (1 µm or less), and further filtered by proximity to a principal cell soma (0.2 µm or less). Remaining inhibitory puncta were counted on the somas of six calbindin positive, and six calbindin-negative principal cells. Dividing puncta counts on calbindin cells by those on calbindin-negative cells yielded synaptic innervation bias measurements such that counts from 12 cells are used to generate a single data point. A value less than one signifies an avoidance of calbindin-positive targets and numbers greater than one signifies a preference for calbindin-positive targets. The five points in each group originate from five different slices. Each slice is a ratio of inhibitory puncta on six negative cells, to six positive cells (12 cells total per point). Slices came from six different animals, three Lis mutant and three wild-type age-matched littermates, spanning three litters.

## Principal cell reconstructions

Slices with biocytin filled cells were fixed (4% PFA and stored at 4°C) and processed for visualization using avidin conjugated dye. Slices were resectioned (50–100 µm) and DAPI stained so cells could be visualized, and their somatic depth could be assessed within the larger hippocampal structure. After staining, slices were imaged, and files were imported to Neurolucida (MBF Bioscience) cell tracing software. Once traced, data sheets were exported for apical dendrite shapes and connectivity profiles for each cell and processed in a custom python script to generate the LRI and ORI measurements later used for morphological clustering. This python script has been provided for use and exploration as a supplemental document to accompany this manuscript.

## Slice preparation

Young adult mice (P20-40) were anesthetized with isoflurane before decapitation. Brains were immediately dissected in dishes of ice-cold dissection ACSF (in mM): 1 CaCl$_2$, 5 MgCl$_2$, 10 glucose, 1.25

$NaH_2PO_4 * H_2O$, 24 $NaHCO_3$, 3.5 KCl, 130 NaCl. ACSF was oxygenated thoroughly for 20mins by bubbling vigorously with 95% $O_2$ and 5% $CO_2$ beforehand. For measurement of cell intrinsic properties whole-cell recordings, mono-synaptic inhibition, and disynaptic inhibition experiments coronal slices were cut (350 µm) using a VT 1200S vibratome from Leica Microsystems. Slices were allowed to recover in an incubation chamber at 35°C in the same solution for 30 min. For oscillation experiments, the same extracellular slicing and recording solutions were used, and pipettes contained extracellular solution. Slices were cut horizontally (450 µm) from more ventral hippocampus, as oscillations were often extremely weak or all together lacking from coronal sections. We verified that similar migratory problems with the late-born calbindin population occurred in ventral hippocampus (*Figure 7B*). Oscillation experiment slices recovered for 15 min at 35°C before being transferred to a custom interface incubation chamber.

## Whole-cell physiology

For electrophysiological recordings slices were transferred to an upright Olympus microscope (BX51WI) with a heated chamber (32°C, Warner Inst.) and custom pressurized perfusion system (~2.5 mL/min). Recording ACSF contained the following (in mM): 2.5 $CaCl_2$, 1.5 $MgCl_2$, 10 glucose, 1.25 $NaH_2PO_4 * H_2O$, 24 $NaHCO_3$, 3.5 KCl, 130 NaCl. Electrodes of 4–6 MOhm resistance (borosilicate glass, World Precision Instruments, no. TW150F-3) were prepared on Narishige (PP-830) vertical pipette pullers. Recording were collected using a Multiclamp 700B amplifier (Molecular Devices) with a Bessel filter at 3 kHz and Digitized at 20 kHz using a Digidata 1440A (Molecular Devices). Protocols were designed, executed and analyzed using the pClamp 10.4 software package (Molecular Devices). Liquid junction potentials were not corrected for and series resistance compensation was not applied. Series resistance was monitored throughout experiments using a −5 mV pulse at the start of each sweep and ranged from 12 to 32 MOhms, cells that varied by greater than 30% over the recording were not considered for analysis. Cells were biased to −70 mV in current clamp mode, and held at −70, −30, and +10 mV in voltage clamp mode depending on the requirements of the experiment. For basic properties and morphological recoveries, electrodes were filled with the following, in (mM): 130 K-glu, 0.6 EGTA, 10 HEPES, 2 MgATP, 0.3 NaGTP, 10 KCl. For monosynaptic inhibition experiments, eIPSCs were recorded at −70 mV using electrodes were filled with (in mM): 100 K-glu, 45 KCl, 3 MgCl, 2 $Na_2ATP$, 0.3 NaGTP, 10 HEPES, 0.6 EGTA; yielding an $E_{cl}$ of −27 mV. eIPSCs were evoked by local stimulation for 5–10 min until a stable baseline was established, then omega-conotoxin GVIA (1 µM) was applied while eIPSCs were monitored for changes in amplitude. Similar experiments were performed washing in omega-agatoxin IVA (250 nM), with QX-314 (2 mM) added to the internal solution. Series resistances for conotoxin: Wt: complex base 20.1 ± 2 wash 23.2 ± 2.9, simple base 18.5 ± 3.8 wash 21.8 ± 4.5 MOhms. Lis1: complex base 18.4 ± 3 wash 21.2 ± 4.5, simple base 17.4 ± 2.3 wash 19.9 ± 2.6 MOhms. Series resistances for agatoxin: Wt: complex base 20.8 ± 1.8 wash 24.7 ± 2.4, simple base 18 ± 1.3 wash 20.5 ± 1.9 MOhms. Lis1: complex base 22.7 ± 1.8 wash 27.8 ± 1.9, simple base 21.6 ± 1.4 wash 25.6 ± 2.4 MOhms. For feedforward I/E experiments electrodes contained (in mM): 135 Cs-MethaneSO_4, 5 NaCl, 4 MgATP, 0.3 NaATP, 10 HEPES, 0.6 EGTA, 5 QX-314 chloride salt, giving an $E_{cl}$ of −69.7 mV. Internal solutions were adjusted for a pH of 7.4 using KOH and an osmolarity of 290 mOsm. Biocytin (2 mg/1 mL) was added to thawed aliquots before use. For feedforward inhibition experiments, pilot experiments where stimulation was delivered in CA3 did not include a wash-in of excitatory blockers as activation of direct monosynaptic inhibition was less likely. For most of the experiments, however, stimulation was delivered in the s. radiatum of CA1 and APV (50 uM)/DNQX (20 uM) was added to block glutamatergic transmission, permitting us to determine and subsequently subtract the monosynaptic component of the inhibitory response. These data were pooled. Experiments where IPSCs were not reduced by at least 30% by wash-in were excluded.

## Extracellular field potentials

For LFP recordings, slices were transferred onto an interface chamber with two manipulator-controlled electrodes positioned under 25x visual guidance. Carbachol (20 µM) was applied to induce slice oscillations. Recordings were made at 10 kHz, low and high pass filtered (8 and 100 Hz, respectively) and mean subtracted. Cross correlation was the max real value resulting from the inverse fast-fourier transformation of $F_1$ and $F_2$; where $F_1$ = fft(signal sample from channel 1), and $F_2$ is likewise

for channel 2, after a flip operation. Cross correlation summary values are the max cross-correlation value in the resulting vector C. The temporal shift between the two signals is the X-coordinate (in milliseconds), corresponding to this cross-correlation peak. Experiments were processed such that channel-1 and channel-2 always corresponded to the same side of the principal cell layer (deep vs superficial).

## Data analysis

Initial data exploration and analysis was performed in custom Python scripts. For further plotting and statistical analysis, Graphpad Prism was used for physiological data. For soma positioning measurements and gephyrin puncta quantification, Microsoft excel sheets were used. Pre-clustering analysis was carried out in python or R using nbclust. K-means clustering was performed in Python using the Scikit learn clustering and decomposition packages. Both clustering routines were supervised (*Figures 3* and *4*), in that they expected K-means n = 2. For morphological clustering this was to replicate prior work and aid in identification of calbindin positive and negative principal cells. For physiological properties, we wished to ask if the two morphological populations might be reflected in our physiology data.

## Statistics

P values represent Welch's t-tests for comparisons of two independent samples, unless otherwise noted. Student's paired t-tests were used for intra-sample (like inhibitory puncta) and pre-post wash comparisons. R values represent Pearson's cross-correlation unless otherwise noted. Quantification and error bars are standard error of the mean.

## Acknowledgements

JD'Amour is supported by the National Institutes of General Medical Sciences (NIGMS) Postdoctoral Research Associate (PRAT) fellowship, award number Fi2 GM123992. C McBain is supported by the *Eunice Kennedy Shriver* National Institute of Child Health and Human Development. The authors acknowledge with gratitude S Hunt and D Abebe for their assistance in tissue processing and animal management, G Akgul for designing new primers to genotype the recombined *Pafafh1b1* allele, M Craig for help with analysis of oscillation experiments, D Calvigioni for assistance with *Neurog2-cre* experiments, C Bengtsson-Gonzales for analysis help and suggestions, and K Pelkey, R Chittajallu, T Petros, S Lee, W Lu. for feedback, comments, suggestions, and discussions during lab meetings. Finally, we are thankful to Dr. Wynshaw-Boris and his lab for providing the heterozygous floxed *Pafafh1b1* mouse used to rederive the full het animal used here.

## Additional information

### Funding

| Funder | Grant reference number | Author |
|---|---|---|
| National Institute of General Medical Sciences | Fi2 GM123992 | James A D'Amour |
| Eunice Kennedy Shriver National Institute of Child Health and Human Development | | Chris J McBain |

The funders had no role in study design, data collection and interpretation, or the decision to submit the work for publication.

### Author contributions

James A D'Amour, Conceptualization, Formal analysis, Funding acquisition, Investigation, Methodology, Writing - original draft, Writing - review and editing; Tyler Ekins, Data curation, Formal analysis, Investigation, Methodology; Stuti Ganatra, Xiaoqing Yuan, Data curation, Investigation, Methodology; Chris J McBain, Conceptualization, Resources, Data curation, Formal analysis, Supervision, Validation, Investigation, Writing - original draft, Writing - review and editing

## Author ORCIDs

James A D'Amour (iD) https://orcid.org/0000-0002-8144-3692
Chris J McBain (iD) https://orcid.org/0000-0002-5909-0157

## Ethics

Animal experimentation: All experiments were conducted in accordance with animal protocols approved by the National Institutes of Health Animal Care and Use Committee (protocol 11-045). All practices aligned with the recommendations of the American Veterinary Medical Association. Care was taken to minimize any suffering.

## Decision letter and Author response

Decision letter https://doi.org/10.7554/eLife.55173.sa1
Author response https://doi.org/10.7554/eLife.55173.sa2

# Additional files

## Supplementary files

- Source code 1. Code used to cluster and sort cellular morphologies.

- Transparent reporting form

## Data availability

All data generated or analysed during this study are included in the manuscript and supporting files. Source data files have been provided for all figures.

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
