## [Decision Letter]

**Acceptance summary:**

In the present work, the authors examined cellular development and network formation in hippocampal CA1 area under in a mouse model of Lissencephaly, featuring severe cellular heterotopia, with a subpopulation of calbindin-expressing principal cells showing inverted laminar positioning. While the misplace cells retained much of their morphological and intrinsic properties, the authors detected very specific deficits between synapses of later born cholecystokinin-expressing interneurons and ectopic calbindin-expressing principal cells, leading to network hyperexcitability and cross-laminar desynchronization. This work demonstrates that layering may play an instructive role in synaptic specification and identified specific circuit motifs that are more susceptible to disruption.

**Decision letter after peer review:**

Thank you for sending your article entitled "Aberrant sorting of hippocampal complex pyramidal cells in Type I Lissencephaly alters topological innervation" for peer review at *eLife*. Your article is being evaluated by two peer reviewers, and the evaluation is being overseen by a Reviewing Editor and John Huguenard as the Senior Editor.

Summary:

The manuscript report a study on the effect of cellular heterotopia on the morphology, connectivity and electrical properties of pyramidal neurons in the pyramidal cell layer of the hippocampus. The authors take advantage of the *Lis^+/-^* model of lissencephaly to investigate how mislamination may affect neuronal properties. The *Lis^+/-^* mouse reports evidence for impaired migration of hippocampal calbindin expressing pyramidal neurons. The mislamination appears to depend on the time of birth of calbindin neurons during embryonic development. Mislamination does not appear to substantially alter the morphology of the neurons, has subtle effects on electrical properties, but appears to have more substantial effects on inhibitory innervation by CCK expressing inhibitory neurons. Additional differences in oscillatory activity induced by carbachol is reported. The authors conclude that mislamination is not a major driver of functional defects, but a specific circuit motif, the CCK-pyramidal motif, is primarily affected by the *Lis^+/-^* mutation.

While the authors find some interesting differences between control and Lis1 mutants, there are no experiments supporting the central claim that the PC neurons analyzed are indeed calbindin cells and unfortunately many of the results suffer from circular inference. The major issues that need to be addressed are detailed below.

Essential revisions:

1) From the Abstract all the way to the Discussion it is rather hard to identify a coherent view of what is the main question the authors are trying to address. It is unclear if their focus is on lamination (as stated initially), or morphological alterations, or circuit connectivity. All these points are addressed, but the rationale guiding the experimental design and description is for the most part understandable only by the intuition of the reader. A clearer and more explicit description of the goals and logic would make the manuscript appealing to a broader audience.

2) Figure 3: The authors start by explaining that "calbindin-expressing principal cells have more complex apical dendritic trees (more branching), than calbindin-negative counterparts". The claim is based on past literature but the authors never experimentally test this claim. In fact, the authors perform supervised clustering (assuming K-means n = 2, due to literature "complex" vs. "simple" trees) and then assume for the remaining manuscript that the "cluster complex" corresponds to "calbindin-cells". This poses several major issues. First, the literature never demonstrated the existence of 2 clusters in Lis1 mice, thus it is not known whether PC with "complex apical trees" correspond to calbindin-cells in these mice, or if there are even 2 clusters, or 3, or 1 cluster in Lis1 mice. Second, the authors cluster cells based on "complexity" and then parameters such as "apical bifurcation" and "sholl complexity" (two parameters inherent to the clustering algorithm) are used to compare cells between the "simple" vs. "complex" cluster, an example of circular inference. Third, the authors show in Figure 1 that calbindin cells represent about 25% of cells in WT and 4% in Lis1, and then according to the algorithm, 54% of cells in WT and 38% in Lis1 are complex, further suggesting that the clustering assumption of "complex cluster of cells" equals "calbindin cluster of cells" is likely incorrect.

3) Figure 4: N numbers are again different across the different panels but are supposed to represent ephys parameters extracted from the same recorded patch cell. In Figure 4C, the algorithm mis-categorized 2 in 8 cells (25%) and 3 in 11 cells (27%), showing unacceptable performance for a clustering algorithm. Another concern is the fact that 8 different parameters were used for physiological clustering analysis (resting membrane potential, sag index, input resistance, spike amplitude, adaptation ratio, firing frequency at 2x threshold, spike threshold, and after hyperpolarization amplitude) but no cross-validation was performed. Risk of possible data overfitting is a major concern.

4a) Figure 6: The major concern is related to the validity of the data. Materials and methods indicate that "Series resistance was monitored throughout experiments using a -5mV pulse at the start of each sweep and ranged from 12-32MOhms". Apart from "12-32MOhms" being a too wide range, what was the average value of series resistance per group? What was the % change of series resistance throughout recordings, namely before and after drug applications? Without that information it is not possible to interpret changes in signal amplitude. Lastly, it is not possible to extract conclusions from n=5 vs. n=13 cells, or from n=3 vs. n=7 cells. How many mice were used for this particular experiment?

4b) The sample traces shown in Figure 6 do not reflect the population data. For experiments in Figure 6, the correct reference is Heft and Jonas, 2005, and not Wilson et al., 2001 (which does not address the cell type specificity of the effects of conotoxin and agatoxin on release, but only heterogeneity of the effect by CB1 expressing or lacking pyramidal neurons).

5) Some of the results appear to be primarily driven by one data point or two. The use of estimation statistics would be helpful in determining which parameters are actually changed in the population data (e.g. Figure 5H, Figure 6D and I or in Figure 7E, or Figure 8D and E).

6) It appears that the size of the hippocampus differs between control and mutant mice (Figure 1A). Nevertheless, without scale bars and clarification of the anteroposterior position at which the slices were cut, it is hard to guess whether the photos were taken from comparable portions of the hippocampus.

---

## [Author Response]

Essential revisions:1) From the Abstract all the way to the Discussion it is rather hard to identify a coherent view of what is the main question the authors are trying to address. It is unclear if their focus is on lamination (as stated initially), or morphological alterations, or circuit connectivity. All these points are addressed, but the rationale guiding the experimental design and description is for the most part understandable only by the intuition of the reader. A clearer and more explicit description of the goals and logic would make the manuscript appealing to a broader audience.

We have added and revised the Introduction and Discussion to focus the reader’s attention to the roles layers play in the developing microcircuitry.

2) Figure 3: The authors start by explaining that "calbindin-expressing principal cells have more complex apical dendritic trees (more branching), than calbindin-negative counterparts". The claim is based on past literature but the authors never experimentally test this claim. In fact, the authors perform supervised clustering (assuming K-means n = 2, due to literature "complex" vs. "simple" trees) and then assume for the remaining manuscript that the "cluster complex" corresponds to "calbindin-cells". This poses several major issues. First, the literature never demonstrated the existence of 2 clusters in Lis1 mice, thus it is not known whether PC with "complex apical trees" correspond to calbindin-cells in these mice, or if there are even 2 clusters, or 3, or 1 cluster in Lis1 mice.

The reviewer brings up a valid concern here that we have also wrestled with. We have included some of the text regarding cluster centers, and birth-dating experiments to the relevant portion of the manuscript. More directly we performed recordings from Lis1 mutants crossed to Calbindin-cre:Ai14 mice, allowing us to make morphological reconstructions from confirmed calbindin expressing principal cells. This line breeds very poorly but we were able to gather 11 recordings, 8 of which had sufficient morphological recoveries to reconstruct and sort algorithmically. Their corresponding LRI and ORI values have been plotted over data in Figure 3B as stars – wherein open stars were confirmed calbindin negative, and filled stars are calbindin positive recoveries – three of these are highlighted in the new bottom panels Figure 3 I, J, K. Note that 4/4 calbindin positive recoveries fall in the upper right “complex” quadrant and would cluster as such, while 4/4 calbindin negative are in the “simple” cluster. An additional 3 recordings were also notably calbindin positive and showed early and heavy apical branching, but the recoveries were incomplete or split across multiple recovered sections hindering processing by the algorithm.

Though as we mention above, the purpose of the algorithm is to be used as a tool to separate two fundamental PC subtypes – we also performed preliminary optimal cluster analysis for morphological data in the form of elbow plots corresponding to the data in Figure 3B. In Author response image 1 show the reduction in the total within-cluster sum of squares as cluster number is stepped from k = 1 to k = 10. Both plots display strong elbows at k = 2.

**Author response image 1. respfig1:** Pre-clustering analysis for morphological and physiological supervised clustering. (**A**) An elbow plot of the total within-cluster sum of squares over various supervised clustering values of k, for morphological LRI and ORI values in Figure 3B. (**B**) Likewise for Lis1 mutant data. (**C**) and (**D**) display summary histograms generated from nbclust, the plots suggest that the optimal cluster answer is k = 2 for both mutant and non-mutant data. (**E**) An elbow plot of the total within-cluster sum of squares over various supervised clustering values of k, for the physiological properties analyzed in Figure 4C. (**F**) Likewise, for Lis1 mutant data in Figure 4D.

Second, the authors cluster cells based on "complexity" and then parameters such as "apical bifurcation" and "sholl complexity" (two parameters inherent to the clustering algorithm) are used to compare cells between the "simple" vs. "complex" cluster, an example of circular inference. Third, the authors show in Figure 1 that calbindin cells represent about 25% of cells in WT and 4% in Lis1, and then according to the algorithm, 54% of cells in WT and 38% in Lis1 are complex, further suggesting that the clustering assumption of "complex cluster of cells" equals "calbindin cluster of cells" is likely incorrect.

Clustering is performed on the generated values for each morphological recovery of ORI (node ratio index) and LRI (length ratio index). These being morphological measures they are related to apical bifurcation and sholl values but are distinct measures. Specifically, because ORI and LRI are examining the “evenness” of the sub- branches arising from any given node with respect to the entire apical dendrite. Regardless, the clustering algorithm has no direct knowledge of what we judge to be the primary apical bifurcation, the sholl values, nor soma positioning. The reviewer takes this as a circular inference, but it is actually shown to lend support to the notion that clustering is working as one would expect. In fact, when we take the labels assigned in clustering and then look at other physical measures of morphology, things make sense. Complex cells tend to have earlier apical bifurcation points and when separated and compared in sholl, they show slightly elevated sholl interactions (our data look very similar to the sholl plots from the original referenced paper as well, where-in it showed the separated PC classes had differences in sholl interactions in the apical tree). This is also the case with the soma positioning, where complex and simple cell somas are superficial and deep in WT animals respectively, while in mutants complex cells are scattered, but substantially deeper on average, as expected from calbindin staining in Figure 1 and the birth dating experiments in Figure 2. The figure also delivers the point that the algorithm is useful. We have found that traditional measures, like sholl, or unidimensional measures like apical bifurcation alone make separating classes difficult. We have supplemented the text to make this clearer.

3) Figure 4: N numbers are again different across the different panels but are supposed to represent ephys parameters extracted from the same recorded patch cell. In Figure 4C, the algorithm mis-categorized 2 in 8 cells (25%) and 3 in 11 cells (27%), showing unacceptable performance for a clustering algorithm. Another concern is the fact that 8 different parameters were used for physiological clustering analysis [resting membrane potential, sag index, input resistance, spike amplitude, adaptation ratio, firing frequency at 2x threshold, spike threshold, and after hyperpolarization amplitude] but no cross-validation was performed. Risk of possible data overfitting is a major concern.

The N’s in Figure 4 vary due to not having recorded each electrophysiological property in every cellular morphology that is reconstructed. Some experiments generate great morphological recovery and no useful recording, sometimes vice versa. Furthermore, some morphologically clustered cells come from later experiments (feedforward inhibition) and therefore will not have corresponding basic physiological properties associated with the data set as different data was collected for these experiments. We have added this to the text accompanying this figure to make it clear.

Regarding the algorithm performance in Figure 4C, I’m worried there is a misunderstanding of what is going on in the figure. We are performing physiological clustering, and comparing that data, with the overlaid morphological clustering labels generated in Figure 3 – (i.e. how well do morphological clusters map on to or represent physiological clusters). This performed above chance in WT mice, and poorly in Lis1 mutants. We present no argument that the clustering is great nor do we argue that it should be great. It is simply an observation about the state of morphological groups loosely reflecting physiological groups and that this relationship is defunct in Lis1 mutants – which speaks to the relative preservation or loss of cellular identity under heterotopia, cell morphology is less directly predictive of physiological properties in mutants.

We also have included (Author response image 1) new preliminary optimal cluster analysis for the eight physiological properties used in Figure 4. Author response image 1 show summary histogram plots generated from nbclust using the hierarchical clustering method – note that for Wt and mutant data, the greatest number of indices point to two clusters as the optimal solution. In addition, Author response image 1 show elbow plots for step increases in cluster number and the corresponding within cluster sum of squares – both Wt and mutant data show elbows at k = 2, further suggesting it is not an unreasonable supervised cluster number.

4a) Figure 6: The major concern is related to the validity of the data. Materials and methods indicate that "Series resistance was monitored throughout experiments using a -5mV pulse at the start of each sweep and ranged from 12-32MOhms". Apart from "12-32MOhms" being a too wide range, what was the average value of series resistance per group? What was the % change of series resistance throughout recordings, namely before and after drug applications? Without that information it is not possible to interpret changes in signal amplitude. Lastly, it is not possible to extract conclusions from n=5 vs. n=13 cells, or from n=3 vs. n=7 cells. How many mice were used for this particular experiment?

We have included additional methodological clarification, series resistances for monosynaptic inhibition experiments and their wash-ins and performed additional recordings in Lis1 mutants to bolster that data set. Animal numbers vary between experiments but are typically greater than 3 per group for a genotype – however we cannot be sure of the morphological group of any recorded cell beforehand, or the success of the recovery itself, and so some groups had low N here which is largely alleviated now.

4b) The sample traces shown in Figure 6 do not reflect the population data. For experiments in Figure 6, the correct reference is Heft and Jonas, 2005, and not Wilson et al., 2001 (which does not address the cell type specificity of the effects of conotoxin and agatoxin on release, but only heterogeneity of the effect by CB1 expressing or lacking pyramidal neurons).

We have corrected the citation, thank you for catching this. With regard to traces in Figure 6, we have discussed this issue in the 3rd response to concern number 2. Briefly, Lis1 data can be grouped by position or cell identity. The bar plots are showing cell identity plots (complex ~ calbindin expressing), and therefore we are discarding their positions. But as you can see from Figure 1, some calbindin cells remain in the superficial band, albeit a minor fraction. Recordings from this band, that were from complex cells are grouped in the bar plots, meaning they contain normotopic calbindin cells, and ectopic calbindin cells – the deficit we describe with CCK innervation seems most prominent in the ectopic calbindin expressing neurons – hence it appears more bimodal. An explanation of data heterogeneity regarding variations in banding severity and cellular location has been added to the manuscript text.

5) Some of the results appear to be primarily driven by one data point or two. The use of estimation statistics would be helpful in determining which parameters are actually changed in the population data (e.g. Figure 5H, Figure 6D and I or in Figure 7E, or Figure 8D and E).

Estimation of N's needed for statistical power is carried out beforehand, and while some points may vary more than others, the averages and distributions are statistically significant, see the p values, demonstrating that the data are not driven by a single point.

6) It appears that the size of the hippocampus differs between control and mutant mice (Figure 1A). Nevertheless, without scale bars and clarification of the anteroposterior position at which the slices were cut, it is hard to guess whether the photos were taken from comparable portions of the hippocampus.

Scale bars were added throughout. The sizes of hippocampi do indeed vary and we go to great lengths to match them for approximate levels of hippocampus as we can.